

# Dynamic interaction of lakes, climate and vegetation over northern Africa during the mid-Holocene

Nora Farina Specht[1], Claussen Martin[1,2], and Thomas Kleinen[1]

[1]Max Planck Institute for Meteorology, Bundesstrasse 53, 20146 Hamburg, Germany
[2]Meteorological Institute, Centrum für Erdsystemforschung und Nachhaltigkeit (CEN), Universität Hamburg, Bundesstrasse 55, 20146 Hamburg, Germany

**Correspondence:** Nora Farina Specht (nora-farina.specht@mpimet.mpg.de)

**Abstract.**

During the early to mid-Holocene, about $11,500$ to $5,500$ years ago, lakes expanded across the Sahel and Sahara in response to enhanced summer monsoon precipitation. To investigate the effect of these lakes on the West African summer monsoon, previous simulation studies prescribed mid-Holocene lakes from reconstructions. By prescribing mid-Holocene lakes, however, the terrestrial water balance is inconsistent with the size of the lakes. In order to close the terrestrial water cycle, we construct a dynamic endorheic lake (DEL) model and implement it into the atmosphere-land model ICON-JSBACH4. For the first time, this allows us to investigate the dynamic interaction between climate, lakes and vegetation over northern Africa. Additionally, we investigate the effect of lake depth changes on the mid-Holocene precipitation, which was neglected in previous simulation studies.

A pre-industrial control simulation shows that the DEL model realistically simulates the lake extent over northern Africa. Only in the Ahnet and Chotts Basin is the lake area slightly overestimated, which likely relates to the coarse resolution of the simulations. The mid-Holocene simulations reveal that both, the lake expansion and the vegetation expansion cause a precipitation increase over northern Africa. The sum of these individual contributions on the precipitation is however larger than the combined effect, when lake and vegetation dynamics interact. Thus, the lake-vegetation interaction causes a relative drying response over the entire Sahel. The main reason for this drying response is that the simulated vegetation expansion cools the land surface more strongly than the lake expansion, mainly the expansion of the Lake Chad area. The resulting warming response over the larger lake area causes local changes in meridional surface-temperature gradient that decelerate the inland moisture transport from the tropical Atlantic into the Sahel, which causes a drying response in the Sahel. An idealized mid-Holocene experiment shows that a similar drying response is induced when the depth of Lake Chad is decreased by about 1-5 m, without changing the horizontal lake area. By reducing the depth of Lake Chad the heat storage capacity of the lake decreases and the lakes warms faster during the summer months. Thus, in the ICON-JSBACH4 model, the lake depth significantly influences the simulated surface temperature and the simulated meridonal surface-temperature gradient between the simulated lakes and vegetation and thus, the mid-Holocene precipitation over northern Africa.



## 1 Introduction

Climate and environment reconstructions from mid-Holocene sediments show that about $11,500$ to $5,500$ years ago, increased monsoon precipitation over northern Africa caused an expansion of vegetation and lakes over the present-day Sahara (Lézine, 2017; Holmes and Hoelzmann, 2017; Hély and Lézine, 2014; Hoelzmann et al., 1998). This 'green Sahara' was characterized by a spatially heterogeneous vegetation distribution, reaching from humid forest taxa to wooded grassland taxa (e.g. Lézine, 2017; Hély and Lézine, 2014). Besides, so-called mega-lakes that covered an area of more than $25,000$ km$^2$ likely existed in the Sahel, such as Mega-lake Chad (Quade et al., 2018; Hoelzmann et al., 1998; Drake et al., 2022) and Mega-lake Timbuktu (Drake et al., 2022). In the Sahara, several smaller but still substantially large lakes primarily formed in regions where the topographic elevation was close to the groundwater level (Holmes and Hoelzmann, 2017; Lézine et al., 2011a; Drake et al., 2022), such as in the Dafur Catchment ($\sim$210 km$^2$ lake area: Pachur and Hoelzmann, 1991) and Fezzan Catchment ($\sim$5,330 km$^2$ lake area: Drake et al., 2018), Chotts Catchment (>620 km$^2$ lake area: Swezey et al., 1999; Coque, 1962; Drake et al., 2022) or Ahnet Catchment (Drake et al., 2022). However, the mid-Holocene lake and vegetation reconstructions are derived from sediment records that are spatially sparse and often temporally discontinuous (Lézine et al., 2011b).

Therefore, the mid-Holocene reconstructions provide only a fragmentary picture of the North African landscape. Particularly, the existence of mid-Holocene mega-lakes over northern Africa (Quade et al., 2018; Drake et al., 2022) and the extent of lakes and wetlands over the western Sahara differs strongly between individual reconstruction studies (Chen et al., 2021; Hoelzmann et al., 1998; Enzel et al., 2017). Accordingly, the prescribed lake and wetland reconstructions used to investigate the influence of mid-Holocene lakes and wetlands on the North African climate differ between individual simulation studies (Li et al., 2023; Specht et al., 2022; Chandan and Peltier, 2020; Krinner et al., 2012; Broström et al., 1998; Coe and Bonan, 1997).

Mid-Holocene simulation studies show that reconstructed mid-Holocene lakes generally cause a precipitation increase over northern Africa. But results differ on whether this precipitation increase is only localized (Coe and Bonan, 1997; Broström et al., 1998; Chandan and Peltier, 2020), or whether lakes cause an area-wide precipitation increase across the Sahel and Sahara (Krinner et al., 2012; Specht et al., 2022; Li et al., 2023). These differences in the precipitation increase might be related to whether vegetation feedback to the lake extent is considered (Krinner et al., 2012; Specht et al., 2022), or whether a static vegetation is prescribed (Coe and Bonan, 1997; Broström et al., 1998; Chandan and Peltier, 2020). For example, a simulation study by Krinner et al. (2012) shows that a reconstructed 'small' lake extent over northern Africa induces a northward extent of the African rain belt by about $1.5\,^\circ$ during the mid-Holocene, when using a dynamic vegetation model (Krinner et al., 2012). In contrast, the same lake extent causes only a marginal precipitation increase over the Sahel and Sahara, in simulation studies where a static mid-Holocene vegetation is prescribed (Broström et al., 1998). Apart from the difference in how the vegetation is treated, previous simulation studies all prescribe the mid-Holocene lake extent over the Sahel and Sahara from reconstructions (Li et al., 2023; Specht et al., 2022; Chandan and Peltier, 2020; Krinner et al., 2012; Broström et al., 1998; Coe and Bonan, 1997).



By prescribing lakes from reconstructions, though, the simulated terrestrial water cycle is not closed. For example, the lake reconstruction prescribed in previous simulations studies might, on average, evaporate more water to the atmosphere than water is supplied to the lakes by discharge generated from precipitation within the corresponding catchments. Therefore, it remains unclear whether the reconstructed lake extent prescribed in previous simulation studies would be sustained under the simulated mid-Holocene climate, particularly when considering that the lake reconstructions are subject to large uncertainties (Li et al., 2023; Specht et al., 2022; Chandan and Peltier, 2020; Krinner et al., 2012; Broström et al., 1998; Coe and Bonan, 1997).

In this study, we close the terrestrial water cycle over northern Africa, by constructing a dynamic lake model and implementing it into the land component JSBACH4 of the ICON Earth system model (ICON-ESM). This allows us to simulate the growth and shrink of Sahelian and Saharan lakes in interaction with the climate and vegetation over northern Africa under a present-day and mid-Holocene climate and, for the first time, to investigate how the dynamic interaction between the atmosphere, lakes and vegetation affects the mid-Holocene climate over northern Africa. In the following, we describe the concept of the dynamic lake model and the setup of the present-day and mid-Holocene simulations.

## 2    Method

To investigate the dynamic interaction between climate, lakes and vegetation during the mid-Holocene, we conduct a pre-industrial control simulation and a set of mid-Holocene experiments. For the simulations, we use the atmosphere model ICON-A (Giorgetta et al., 2018) and the land model JSBACH4 (Schneck et al., 2022; Reick et al., 2021) at ∼160 km horizontal resolution and 47 vertical atmospheric hybrid sigma levels. The atmosphere–land model is forced with climatological 0 kyr BP (1850) and 6 kyr BP orbital parameters (Berger, 1978) and greenhouse gas concentrations (GHGs) (Fortunat Joos, personal communication, 2016; see Bader et al., 2020; Brovkin et al., 2019) for the pre-industrial (pi) and mid-Holocene (mH) simulations, respectively. The pre-industrial (Wieners et al., 2019a) and mid-Holocene (Jungclaus et al., 2019) sea-surface temperatures (SST) and sea-ice concentrations are prescribed from Coupled Model Intercomparison Project 6 (CMIP6) simulations with the Max Planck Institute Earth System Model (MPI-ESM).

Like most Earth system models (Richter and Tokinaga, 2020), the MPI-ESM simulates a too warm SST (>5 K) in the eastern tropical Atlantic (Jungclaus et al., 2013). This promotes a too high precipitation over the Guinea coast (Zhao et al., 2007) and a too southern position of the West African monsoon during the northern hemisphere summer. To simulate a more realistic latitudinal position of the West African summer monsoon, we subtract a monthly climatology of tropical SST biases from the pre-industrial and mid-Holocene MPI-ESM CMIP6 SST. The tropical SST biases are derived from the differences between a historical MPI-ESM CMIP6 simulation (Wieners et al., 2019b) and observation-based Atmospheric Model Intercomparison Project II (AMIP2) SST data (Durack et al., 2022). Finally, the tropical SST biases are smoothed at the 30 °N and 30 °S boundaries to avoid artificial temperatures edges. We only correct the SST in the tropical regions because we assume the influence from the SST biases of higher latitudes to be comparably small.

Furthermore, all simulations are run with a dynamic background albedo scheme as described by Specht et al. (2022). The dynamic background albedo scheme represent changes in the surface albedo due to litter production from vegetation. Vamborg



et al. (2011) show that these background albedo changes substantially increase the mid-Holocene precipitation over northern Africa. This dynamic background albedo is only applied over the northern Africa.

With this experiment setup, we conduct a pre-industrial equilibrium simulation with dynamic vegetation and dynamic lakes (pidVdL). The pre-industrial simulation is run until the vegetation and lakes reach a close-to-equilibrium state. The following 150 simulation years are then used as the averaging period. The average lake extent of the pidVdL simulation is compared to

an observation-based lake map of the HydroLAKES data set (Messager et al., 2016) to evaluate the accuracy of the DEL model in parameterizing the Sahelian and Saharan endorheic lakes.

In addition, a set of mid-Holocene simulations is conducted to investigate the individual and synergistic effects of vegetation and lake feedback over northern Africa. In these mid-Holocene simulations the vegetation and lakes are either prescribed as static boundary conditions derived from the 150-year averages of the pidVdL simulation or by using the dynamic vegetation

and dynamic lake model:

- pidVdL: pre-industrial simulation with dynamic vegetation and dynamic lakes.

- mHdVdL: mid-Holocene simulation with dynamic vegetation and dynamic lakes.

- mHdV: mid-Holocene simulation with dynamic vegetation and prescribed pre-industrial lakes from pidVdL.

- mHdL: mid-Holocene simulation with prescribed pre-industrial vegetation from pidVdL and dynamic lakes.

- mH: mid-Holocene simulation with prescribed pre-industrial vegetation and lakes from pidVdL.

Similar to the pidVdL run, the individual mid-Holocene simulations are run until the lake and/or vegetation extent over northern Africa have reached a close-to-equilibrium state (Fig. A1). To derive robust results, the individual mid-Holocene simulations have an evaluation period of 150 or 200 years, depending on the variability of the lakes and vegetation in the Sahel and Sahara (Fig. A1: white part).

The individual and synergistic effects of the dynamic lakes and dynamic vegetation on the e.g. the precipitation over northern Africa is investigated by applying the following factor analysis:

$$\delta_{net} = \delta_{veg} + \delta_{lake} + \delta_{syn} \tag{1a}$$

$$\delta_{net} = mHdVdL - mH \tag{1b}$$

$$\delta_{veg} = mHdV - mH \tag{1c}$$

$$\delta_{lake} = mHdL - mH \tag{1d}$$

$$\delta_{syn} = mHdVdL + mH - mHdV - mHdL \tag{1e}$$

Where $mHdVdL - mH$ is the net response due to the influence of both, dynamic lakes and vegetation. $\delta_{veg}$ and $\delta_{lake}$ represent the linear, or pure, response to dynamic vegetation and dynamic lakes, respectively. Finally, $\delta_{syn}$ is the non-linear response caused by synergetic effects of dynamic vegetation and dynamic lakes.



Finally, we consider that the simulated dynamic lakes change not only in extent but also in depth. Previous simulation studies exclusively focus on the climate effect from changes in the lake extent, while the climate effect from changes in the lake depth was neglected (Broström et al., 1998; Carrington et al., 2001; Krinner et al., 2012; Chandan and Peltier, 2020; Specht et al., 2022). Therefore, we conduct an additional mid-Holocene simulation with a prescribed pre-industrial lake and vegetation extent, but a constant 10-m lake depth, as used by Specht et al. (2022), to investigate the effect of the lake depth
changes on the mid-Holocene climate over northern Africa:

   – mHL10: prescribed pre-industrial vegetation and prescribed pre-industrial lakes with a constant 10 m lake depth.

   In all simulations, the lake is represented as pure mixed-layer of a given depth and the lake extent and lake depth only vary within the endorheic catchments over northern Africa (Fig. 1: colored outlines). In the results, we will show that understanding the effect of lake depth changes, in particular of Lake Chad, is essential to explain the simulated synergistic effect over northern
Africa seen in the mid-Holocene experiments.

   The effect of reducing the depth of Lake Chad by 1-5 m from a standard model depth to the actually simulated depth of the pre-industrial Lake Chad is given by:

$$\delta_{depth} = mH - mHL10 \tag{2}$$

   In the analysis some changes for $\delta_{syn}$ and $\delta_{depth}$ are only shown as time averages of the rain season, which is from June to
September (JJAS). Since most of the annual precipitation falls over northern Africa during JJAS, the atmospheric response to dynamic lakes and vegetation that cause precipitation changes over northern Africa become most obvious in this season.

### 2.1   Dynamic endorheic lake model concept

The Sahara and Sahel mainly consist of endorheic catchments, which are catchments without an outlet to external waters such as rivers or the ocean (Fig. 1: colored outlines). Within the endorheic catchments, surface runoff and sub-surface drainage flow
down-slope into internal orographic depressions, where lakes expand and shrink depending on the terrestrial water budget of the corresponding catchment (Fig. 1: symbolized by purple dots). This concept of endorheic catchments is used in our study to simulate the dynamic lake extent over northern Africa. In the following, we describe the technical concept of thr endorheic dynamic lake model (DEL model) and how it is embedded in the hydrological discharge model (HD model: Hagemann and Duemenil, 1998; Hagemann and Dümenil, 1997; Hagemann and Gates, 2001), which is part of JSBACH4 (Reick et al., 2021),
the land component of the ICON Earth system model (Jungclaus et al., 2021).

   The HD model is a river routing model that runs on the same resolution as the land component JSBACH4, which is ∼160 km for this study. The HD model includes an overland flow, base flow and river flow (Fig. 2). These flows are calculated based on the linear reservoir concept, which assumes a time-constant retention time $k$ of water in a reservoir and, thus, a linear relation between the reservoir water storage $S(t)$ and the water outflow $Q(t)$ from that reservoir (e.g. Kang et al., 1998):





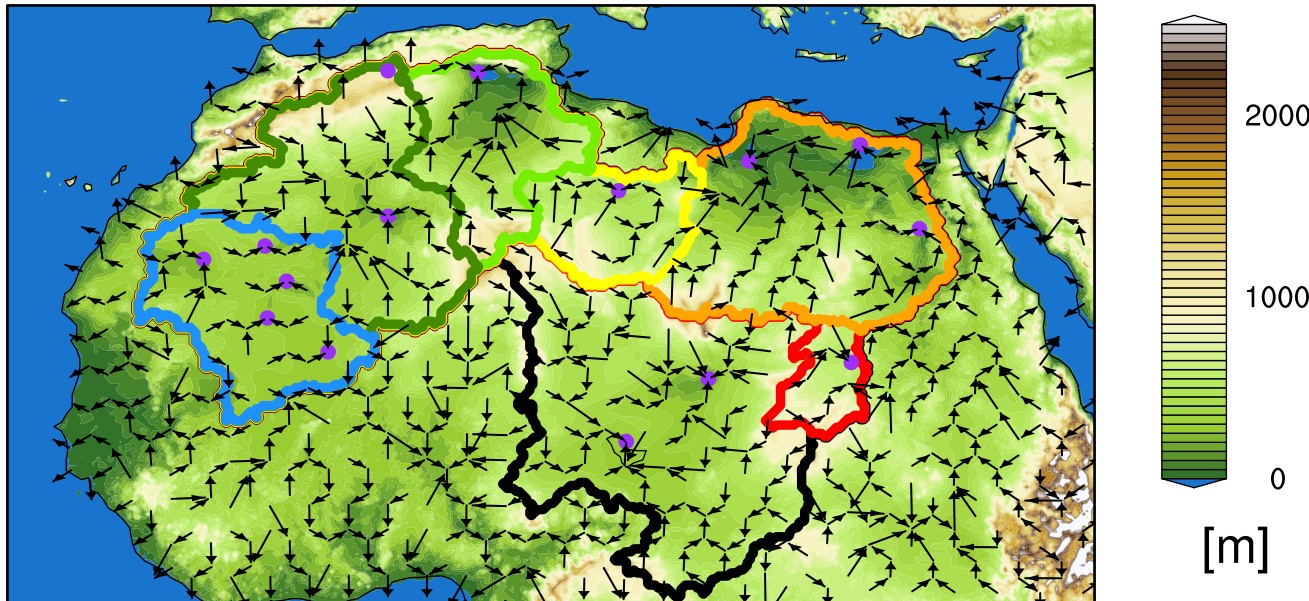

**Figure 1.** River flow directions (arrows) used as boundary conditions to simulate the down-slope water transport within each endorheic catchment (colored outlines) to the internal drainage points (purple dots). The digital elevation model represents the topography in the background. The colored outlines show the Chad Catchment (black), Dafur Catchment (red), Toudenni Catchment (blue), Ahnet Catchment (dark green), Chotts Catchment (light green), Fezzan Catchment (yellow) and a northeast African catchment, respectively. Please note that these endorheic catchment where generate in a way that it suits the resolution of the model simulations, whereas in reality the marked region might be also splinted into smaller sub-catchments.

$$Q(t) = \frac{S(t)}{k} \tag{3}$$


The retention time ($k < 1$ day) for the overland flow and the river flow of the HD model is derived from the local slope of the orography, while the retention time for the slower base flow is set to a constant value of 300 days.

The water inflow to the HD model is given by the surface runoff for the overland reservoir and by the sub-surface drainage for the base reservoir (Fig. 2). The k-dependent outflow from the overland and base reservoir is transported to the river reservoir of the neighboring downstream grid cell (Fig. 2). The river reservoir is a cascade of 5 linear sub-reservoirs that all have the same retention time (Fig. 2). After the water has passed the linear river reservoir cascade, the outflow from this reservoir is transported further downstream to the river reservoir of the next neighboring grid cell. In this way, the discharge water flows down-slope until it reaches either an internal drainage grid cell (Fig. 1: purple dots) or a coastal grid cell.




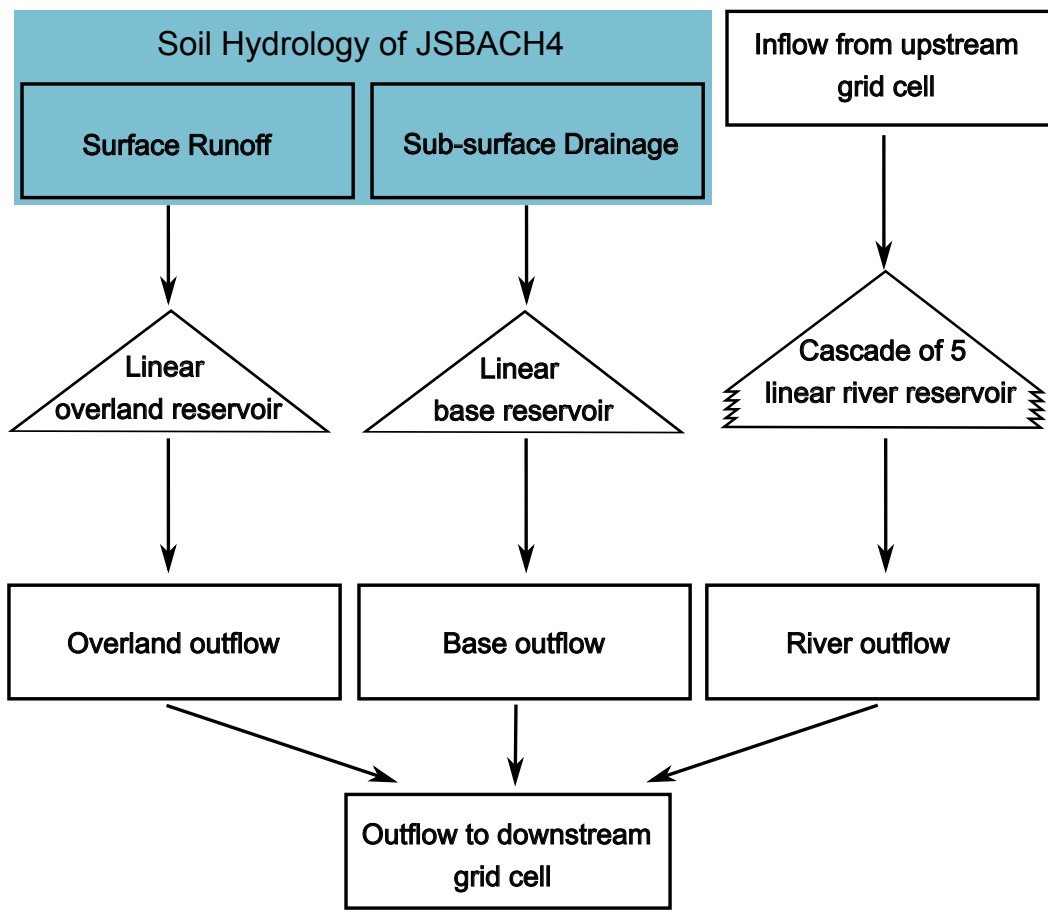

**Figure 2.** Structure of the HD model and inputs from the soil hydrology in blue (adapted from Hagemann and Duemenil, 1998).

At the internal drainage grid cells, discharge water enters the local lake reservoir of the DEL model (Fig. 3). Remote sensing
studies show that, in the Sahel and Sahara, most of the discharge water is transported to the groundwater reservoir below the
internal depressions by a sub-surface flow. E.g. in the Chad catchment, about ~70% of the seasonal discharge contributes to the
sub-surface groundwater reservoir (and soil moisture) and less than 30% of the seasonal discharge directly flows into surface
storage of the Lake Chad (Pham-Duc et al., 2020). The Quaternary groundwater aquifer and the above surface storage of Lake
Chad exchange water on a much longer timescale, i.e. a time period of about 20–40 years is needed to completely replace
the water volume of the Chad lake (as before the Sahelian drought in the 1970s) with water from the Quaternary groundwater
aquifer (Bouchez et al., 2016). In the HD model, such a groundwater reservoir is missing. First simulations showed that
unrealistic high inter-annual fluctuations in the lake extent occur when the discharge from the (surface) river reservoir is added
directly to the lake reservoir. To avoid these unrealistic high fluctuations, the outflow from the river reservoir is added to the



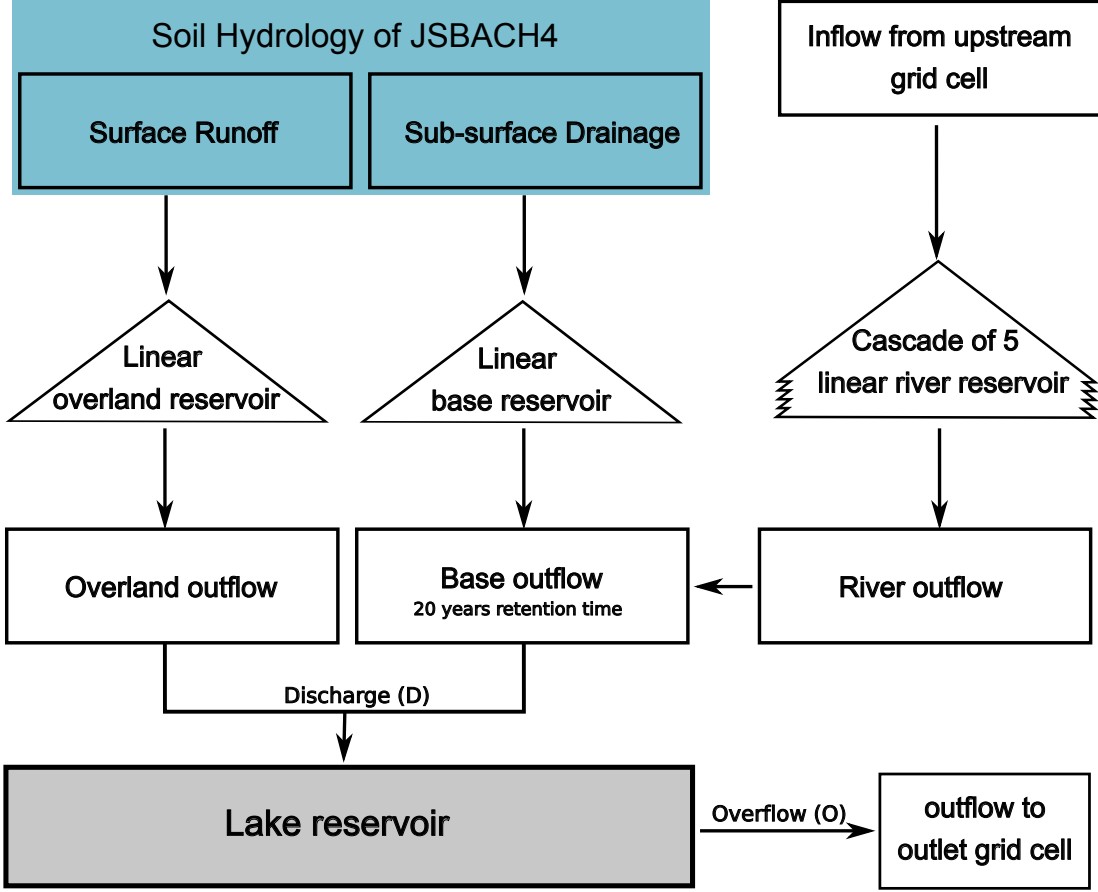

**Figure 3.** Structure of the HD model at the internal drainage point and the embedded endorheic lake reservoir with its inflow from and outflow to the HD model (gray box), same as the inputs from the soil hydrology (blue box) (adapted from Hagemann and Duemenil, 1998).

base reservoir at the internal drainage grid cells and the retention time of this base reservoir is set to 20 years (Fig. 3). This
20-year period is taken from an estimate of the residence time of water in the Quaternary groundwater aquifer before it enters
the Lake Chad (Bouchez et al., 2016). We use this 20-year residence time as rough approximation for all internal drainage grid
cells within the Saharan endorheic catchments (see Sec. 2.2).

The lake water volume changes $\Delta V_{lake}$ of the DEL model are described by:

$$\Delta V_{lake} = (D - O + f_{lake}(P - E_{lake}) + Fin_{lake} - Fout_{lake}) \cdot \Delta t \qquad (4)$$



$D$ represents the discharge water at the internal drainage grid cell given by the HD model (Fig. 3). $O$ is the lake water outflow that is returned to the HD model at the outlet point of the respective basin (Fig. 3). This overflow only occurs when the lake level exceeds the outlet height of the lake basin, that is, when the maximum basin volume $V_{lake,max}$ within a grid cell is exceeded (which did not happen in our simulations). $f_{lake}$ represents the lake fractions of the grid cell that directly interact with the atmosphere through precipitation $P$ and evaporation $E_{lake}$. The evaporation over the lake surface is assumed to be

equal to the potential evaporation that depends on the surface temperature and surface roughness of the lake. $Fout_{lake}$ and $Fin_{lake}$ describe the lateral lake water flow between neighboring grid cells of the same lake basin as a lake grows or shrinks. This lateral lake water flow compensates lake level differences between grid cells of the same lake basin or sub-basin.

 The lateral lake water flow between neighboring grid cells depends on the absolute lake level height $H_{dyn}$ of each grid cell, which is the sum of the orographic minimum elevation or bottom point of a grid cell $H_{oro,min}$ and the lake level $H_{lake}$ above

this bottom point:

$$H_{dyn} = H_{oro,min} + H_{lake} \tag{5}$$

 While $H_{oro,min}$ is given by fixed boundary conditions, $H_{lake}$ is derived by using a lookup table that describe the lake volume-area-height relation for each grid cell at 1 % lake fraction intervals see (Sec. 2.3). The lake level and lake area for a given lake volume is derived by linear interpolation between the intervals of this lookup table. The lookup table intervals are

set to 1 % lake fraction intervals to avoid lake fraction errors of >1 % for a given lake volume per ICON grid cell.

 To accurately represent the lake water flow between neighboring grid cells with regard to the basin and sub-basin orography, the maximum ridge elevation between each grid cell and its 3 neighboring lake grid cells ($H_{oro,ridge}$) is required (see Sec. 2.3). $H_{oro,ridge}$ allows the representation of several sub-basins, like the Chad Basin and the Bodélé Depression of the Chad Catchment. E.g. lake water only flows from the present-day Chad lake into the northern Bodélé Depression, when the lake

level of the Chad lake exceeds $H_{oro,ridge}$.

 Based on $H_{dyn}$ and $H_{oro,ridge}$, the dynamic flow directions and the flow velocity between grid cells of the same basin are derived. The dynamic flow directions and the corresponding flow velocity depend on the absolute lake level differences $\Delta H_{dyn}$ between two neighboring lake grid cells:

$$\Delta H_{dyn} = H_{dyn}(up) - H_{dyn}(down) \tag{6}$$

$H_{dyn}(up)$ represents the dynamic lake level of the upstream grid cell and $H_{dyn}(down)$ represents the dynamic lake level of the downstream grid cell. $\Delta H_{dyn}$ is recalculated at each time step.

 The flow velocity is derived by using the Manning-Strickler equation, which describes an open channel flow (Strickler, 1981):

$$v_{lake} = k_{st} \cdot \Delta H_{dyn}{}^{2/3} \cdot \left( \frac{\Delta H_{dyn}}{\Delta x} \right)^{0.5} \tag{7}$$





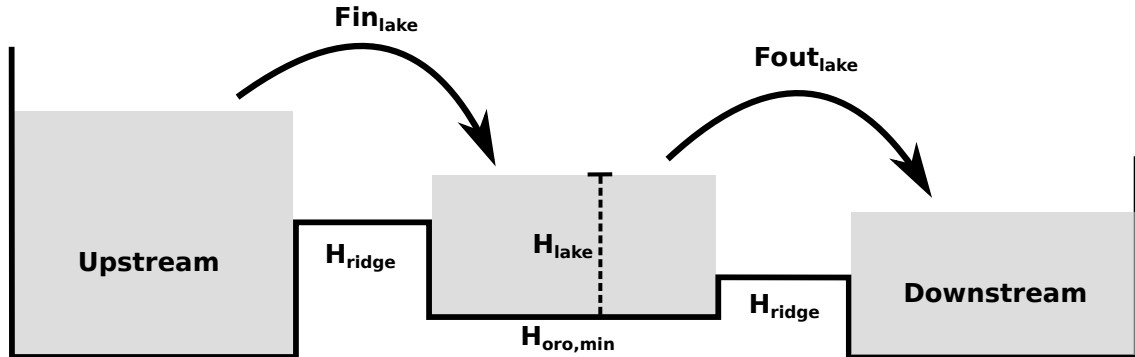

**Figure 4.** Structure of the lateral flow of the DEL model.

$v_{lake}$ is the flow velocity between two neighboring grid cells. $k_{st}$ $[m^{1/3}s^{-1}]$ is the roughness coefficient according to Strickler, which is a measure for the wall roughness of the channel flow (Strickler (1981): page 10-12). $k_{st}$ is set to a constant value of $k_{st} = 100$, which refers to a fast flow over smooth concrete. By setting this high value, we assume that lake level differences between neighboring grid cells are equalized on a short timescale. $\Delta x$ is the distance between the center points of two neighboring grid cells.

The lateral lake flow, $F_{lake}$, represents the lake water inflow $Fin_{lake}$ of the downstream grid cell and the lake water outflow $Fout_{lake}$ of the upstream grid cell in Eq. (4):

$$F_{lake} = \min\left(V_{lake} \cdot \frac{v_{lake}}{\Delta x}, \frac{V_{lake}}{\Delta t}\right) \qquad (8)$$

where $V_{lake}$ is the lake water volume of the upstream grid cell and $\Delta t$ is the integration time step of the model.

     Finally, in JSBACH4 lakes are, by standard, represented by as a mixed layer with a constant depth of 10 m. The simulated

lakes of the DEL model, in contrast, are represented by a mixed layer with a dynamic lake depth. This dynamic lake depth corresponds to the simulated lake level $H_{lake}$ of the current time step, in which a minimum depth of 1 m is applied. The dynamic lake depth affects the energy storage of the lakes and, thus, its surface temperature.

## 2.2    Boundary conditions for the HD model

The HD model requires flow directions and reservoir retention times as input boundary conditions in order to simulate the

down-slope transport of discharge water to internal drainage grid cells or ocean grid cells. The flow directions and reservoir retention times for the HD model are generated using the MPI-DynamicHD model version 1.3 (Riddick et al., 2018). The MPI-DynamicHD model requires a $10'$ (minutes of arc) orography and the location of the internal drainage grid cells flagged on a $10'$ grid as input data. These input data are derived from a $30''$ (seconds of arc) digital elevation model (DEM) and endorheic



catchment outlines of the HydroSHEDS data set (Hydrological data and maps based on SHuttle Elevation Derivatives at multiple Scales) that are regridded to a 10′ resolution (Lehner and Grill, 2013).

The internal drainage grid cells are here defined as bottom points of a lake basin or lake sub-basin. To derive these internal grid cells on a 10′ resolution, the 10′ orography data and the ICON grid at a ∼160 km horizontal resolution are used as input data. In a first step, the 10′ orography the is used to find local minima in a box of 9 x 9 grid cells. The local minimum that has the lowest elevation within the region of each coarser ICON grid cell is selected and flagged as potential internal drainage grid cell. Subsequently, the basins of these flagged local minima are flooded, starting with the local minimum that has the lowest elevation. The orography around the local minimum is flooded until (1) the boundaries of the corresponding endorheic catchment is reached or (2) the next downstream lake basin is reached or (3) the flooding height is below the elevation of the current local minimum.

After all local minima and their basins are flooded, only the largest basin and all smaller downstream basins within each endorheic catchment are selected. Additionally, the flooded basin of a local minimum have to be at least 20 % the area of an ICON grid cell. Since large basins might contain several sub-basins, all local minima within a basin are considered that are at least as large as the area of one ICON grid cell. The above mentioned minimum basin and sub-basin area is set in a way that the MPI-DynamicHD model generated reasonable flow direction on the coarse R2B4 ICON resolution (˜160 km: Fig. 1: black arrows).

After processing the flow directions and the retention times with the MPI-DynamicHD model, the retention times of the base flow reservoirs at the internal drainage grid cells are set to 20 year, as mentioned in section 2.1. This reduces unrealistic high fluctuations in the simulated lake discharge water inflow so that the lake size becomes less oversensitive to the inter-annual precipitation variability.

## 2.3 Boundary conditions for the DEL model

The DEL model requires a set of boundary conditions to simulate the growth and shirk of endorheic lakes over northern Africa. To simulate the lake extent and lake height a lake level-area-volume lookup table for each grid cell ($H_{lake}$ and $f_{lake}$) is needed. To simulate the water flow between neighboring lake grid cells, the minimum geographic height for each grid cell ($H_{oro,min}$) and the minimum ridge height to the 3 neighboring grid cells ($H_{oro,ridge}$) are additionally needed. To simulate the overflow of the lakes in case the whole lake basin is flooded, the maximum lake volume ($V_{lake,max}$) and the lake basin outlet point is required. All these boundary conditions are derived from a 30″ digital elevation model (DEM) and endorheic catchment outlines of the HydroSHEDS data set that are regridded to a 10′ resolution (Lehner and Grill, 2013).

The minimum geographic height ($H_{oro,min}$) for each grid cell is derived by identifying the minimum elevation of the 10′ orography within each coarse resolution ICON grid cell. The lake level-area-volume lookup table is derived by gradually flooding the 10′ orography within the boundaries of each ICON grid cell individually. The minimum ridge height to the 3 neighboring grid cells ($H_{oro,ridge}$) is derived by flooding the 10′ orography starting at the point of minimum elevation of the current ICON grid cell until the point of minimum elevation of the neighboring ICON grid cell is reached. The maximum elevation point of this flooding process is set as minimum ridge height for the corresponding neighboring grid cell.



The maximum lake volume $V_{lake,max}$ and lake basin outlet points, used to simulate the overflow of a lake basin, are derived based on the internal drainage grid cells and $10'$ orography described in Sec. 2.2. The outlet point of each lake basin is derived by flooding the $10'$ orography starting at the internal drainage grid cells. The $10'$ orography is flooded until the first ICON grid cell that lies outside the lake basin is reached. This ICON grid cell is set as outlet grid cell. The maximum lake volume for each grid cell is derived from the different between the outlet height and the $10'$ orography.

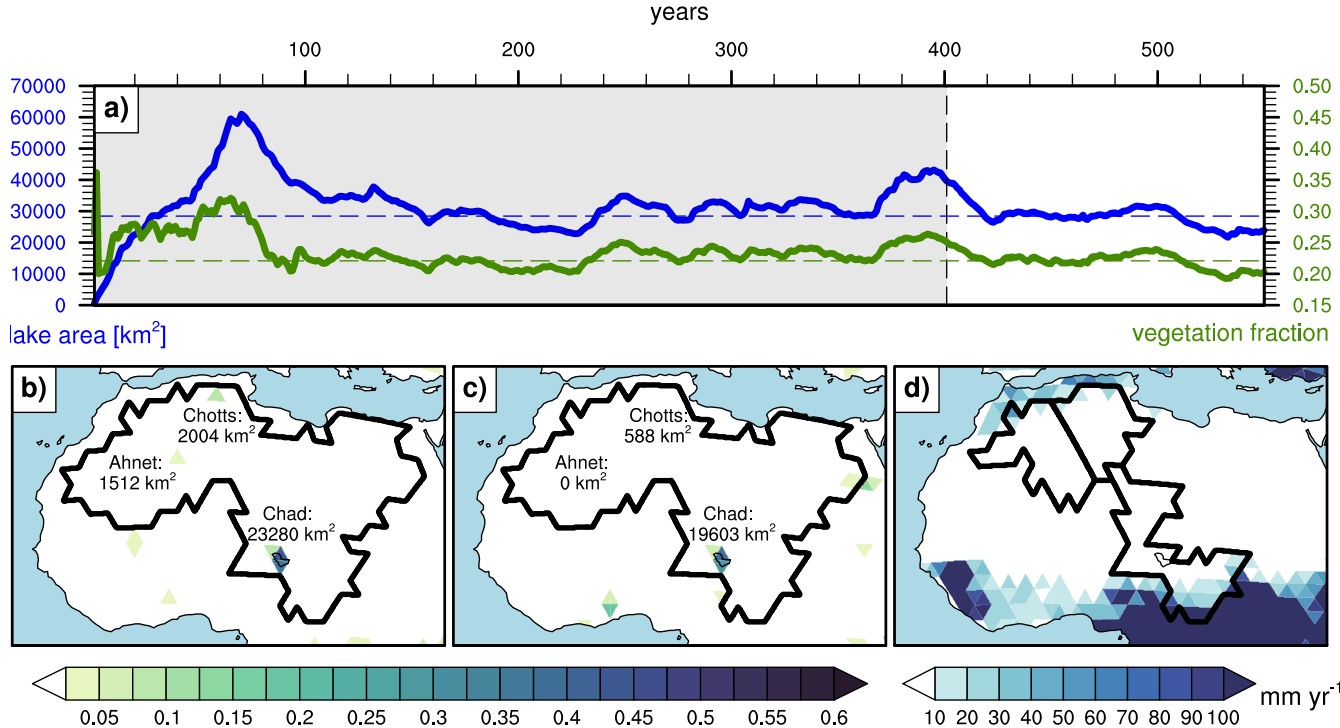

**Figure 5.** The upper plot shows the simulated pre-industrial (a) time series of the lake area withing the endorheic catchments over norther Africa (blue line) and the vegetation cover averaged over northern Africa (green line) (20 °W-35 °E,10 °N-35 °N). The lower panel shows the (b) simulated pre-industrial lake cover fraction averaged over the last 150-year years of pidVdL simulation, (c) the observation-based pre-industrial lake cover fraction derived from the HydroLAKES data (Messager et al., 2016), and (d) the simulated pre-industrial runoff and drainage averaged over the last 150-year years of pidVdL simulation. The black boundaries in subplot b) and c) show the endorheic catchment region, in which the lake extent is simulated dynamically. The black boundary in subplot c) indicates the boundaries of the Ahnet, Chotts and southern Chad Catchment.



## 2.4    Results

The pidVdL simulation is run for 550 year, in which the lake and vegetation cover over northern Africa reach a close-to-
equilibrium state after about 400 years (Fig. 5a: gray part). The last 150 years of the simulation are used as evaluation period
(Fig. 5a: white part).

The simulated surface area of Lake Chad from pidVdL is about $3,677$ km$^2$ ($\sim 18.8\%$) larger than the observation-based
surface area of Lake Chad from the HydroLAKES data (Fig. 5 b-c). A surface deviation of 18.8% is relatively small, since
Lake Chad is characterized by a wide and shallow basin. Accordingly, small changes in the water budget already lead to
comparatively large changes in the surface extent. For example, a drought in the 1970s and 1980s reduced the surface area
of Lake Chad by about 90% (e.g. Olivry et al., 1996), from about $20,000$ km$^2$ (1950-1972) to less than $2,000$ km$^2$ (1980s)
(e.g. Pham-Duc et al., 2020; Bouchez et al., 2016). Considering this large changes in the surface extent, we conclude that the
HD-DEL model properly simulates the pre-industrial equilibrium state of Lake Chad.

The simulated surface area of Lake Ahnet and Lake Chotts from pidVdL is overestimated by about $1,512$ km$^2$ (currently
a dry basin) and $1,416$ km$^2$ ($\sim 340,8\%$) in comparison to the HydroLAKES data (Fig. 5 b-c). Both lakes receive most of
their water inflow through runoff and drainage from the Atlas Mountains in the northwestern region of Africa (Fig. 5 d). The
overestimated water inflow from the Atlas Mountains is likely due to the coarse resolution of the ICON-JSBACH4 model
which causes an inaccurate representation of the catchment's watershed, as well as an inaccurate representation of the small-
scale heterogeneous precipitation from orographic updrafts.

Other simulated lakes from pidVdL, that are surrounded by a flatter terrain, are in better agreement with the HydroLAKES
data (Fig. 5 b-c). This includes lakes in the northwestern Sahara, whose surface extent is less than 2.5% of an ICON grid
cell (Fig. 5b). Similarly, simulated lakes that border mountains that render less orographic-forced precipitation are in better
agreement with the HydroLAKES data. e.g. Lake Chad mainly receives most of its discharge water from the southern part of
the catchment, rather than from the Ahaggar and Tibesti Mountains in the north (Fig. 5 d). Satellite observations show that
>90% of the discharge water of Lake Chad is provided by the Chari/Logone river system in the southern part of the catchment
(E.g. Pham-Duc et al., 2020).

Thus, the HD-DEL model accurately simulates the pre-industrial surface extent of Saharan lakes like Lake Chad, except for
lakes adjacent to mountains that receive most of their discharge water from orographic-forced precipitation.

### 2.4.1    Mid-Holocene precipitation, lake and vegetation changes

Comparison between mHdVdL and pidVdL shows that the prescribed mid-Holocene forcing (orbit, GHGs, SST and SIC)
and the simulated mid-Holocene lakes and vegetation extent, together, cause a dipole like "wet-north-dry-south" precipitation
response around 10 °N (Fig. 6a). This dipole-like precipitation change is associated with a northward shift of the West African
summer monsoon rain belt. Along with the mid-Holocene precipitation increase over the Sahel and western Sahara, Lake Chad
expands at its southern boundary and the overflow of the Chad Basin into the Bodélé Depression causes an additional lake
expansion northeast of Lake Chad (Fig. 6b). Additionally, some smaller lakes also form in the western Sahara (Fig. 6b). The



vegetation cover mainly increases over the Sahel (12 °N-18 °N) and over the western Sahara, similar to the pattern of the precipitation increase over northern Africa (Fig. 6a,c).

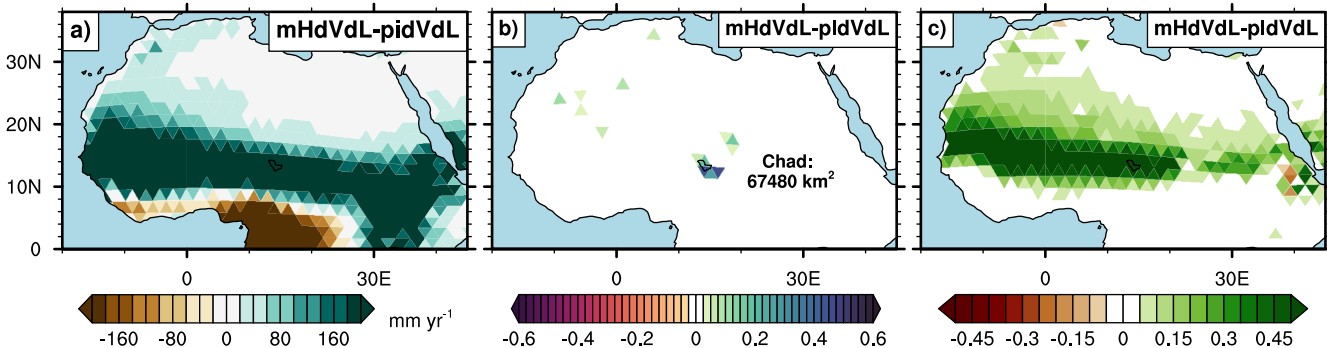

**Figure 6.** Simulated mid-Holocene (a) annual precipitation changes in $mm/yr$, (b) lake fraction changes and (c) vegetation cover changes compared to the pre-industrial simulation (mHdVdL-pidVdL). The lake and vegetation cover fraction describe the cover fraction within each grid cell and, therefore, are unitless.

The simulated mid-Holocene lake and vegetation extent shown in Fig. 6 also occurs when starting the mHdVdL with different initial conditions. A mid-Holocene simulation similar to mHdVdL, but initiated with completely filled lake basins and a vegetation cover of 100% over northern Africa runs into a similar equilibrium state (Fig. A1 a,d). These results suggest, that simulated mid-Holocene lake and vegetation extent in the ICON-JSBACH4 model is independent of the prescribed initial conditions.

The ICON-JSBACH4 model seems to underestimate the mid-Holocene precipitation increase and, thus, the lake and vegetation expansion over northern Africa (Fig. 6). The simulated precipitation in the central Sahara increases by up to 100-200 mm year$^{-1}$ (Fig. 6a: 18 °N-35 °N), whereas reconstructions suggest an increase of about 200-600 mm year$^{-1}$ in this region (Braconnot et al., 2012; Bartlein et al., 2011). The simulated lake extent is also substantially smaller than sediment reconstructions suggest. For, example, the simulated Lake Chad expands up to an area of 67.480 km$^2$ (Fig. 6b), whereas reconstructions suggest a area of about $350,000$ km$^2$ (Hoelzmann et al., 1998; Quade et al., 2018; Drake et al., 2022). Similar to the lakes, the simulated extent of the Sahelian-Saharan vegetation is underestimated as well (Fig. 6c), in comparison to reconstructions (Hély and Lézine, 2014; Lézine, 2017; Hoelzmann et al., 1998) and climate models that produce a more humid climate over northern Africa (e.g. Dallmeyer et al., 2021).

This dry bias over northern Africa is a known issue of the ICON-JSBACH4 model (Schneck et al., 2022) and some other state-of-the-art climate models (Braconnot et al., 2012). Obviously, the presence of a dynamic lake in the the ICON-JSBACH4 model does not diminish the dry bias of the model. In the following we analyse to what extent dynamic lakes contribute to a greener Sahara and we analyze the individual and synergistic contributions of dynamic lakes and a dynamic vegetation to the mid-Holocene precipitation increase.




### 2.4.2 Mid-Holocene precipitation response to dynamic lakes and dynamic vegetation

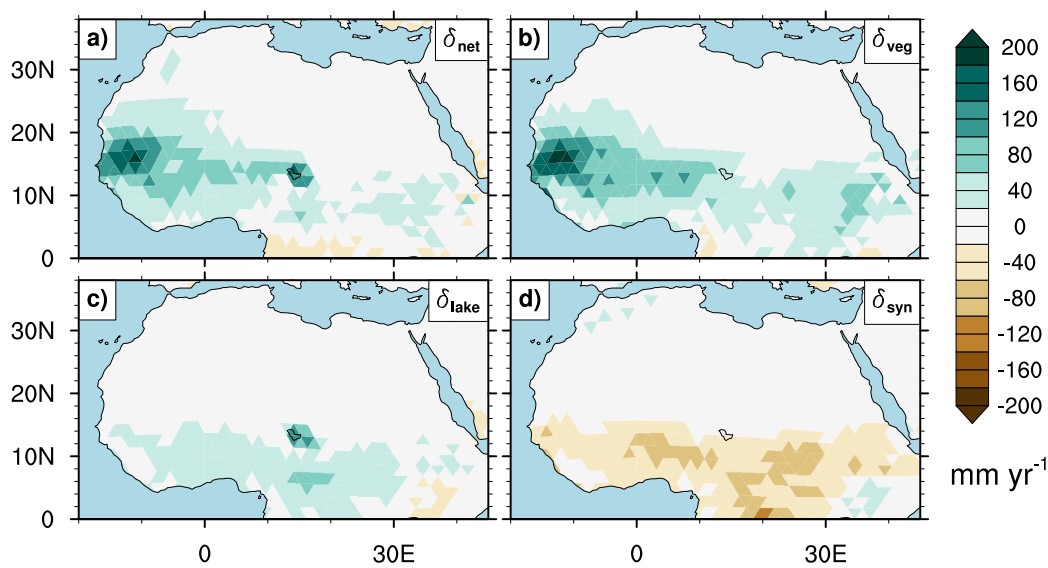

**Figure 7.** Simulated mid-Holocene annual mean precipitation changes caused by (a) dynamic lakes and dynamic vegetation together ($\delta_{net}$: mHdVdL-mH), (b) dynamic vegetation alone ($\delta_{veg}$: mHdV-mH), (c) dynamic lakes alone ($\delta_{lake}$: mHdL-mH), and (d) vegetation-lake synergies ($\delta_{syn}$: mHdVdL+mH-mHdV-mHdL). The mid-Holocene net precipitation changes ($\delta_{net}$: mHdVdL-mH) decompose as $\delta_{net} = \delta_{dV} + \delta_{dL} + \delta_{syn}$.

The simulated mid-Holocene expansion of dynamic lakes and dynamic vegetation causes a northward shift of the West African summer monsoon, with a precipitation decrease south of 5 °N and a precipitation increase north of 5 °N (Fig. 7a).

The precipitation increase is strongest over Lake Chad and over the western Sahel and Sahara (Fig. 7a). This corresponds to the regions where the simulated mid-Holocene lake expansion (Fig. 6b) and vegetation expansion (Fig. 6c) is strongest. The precipitation increase caused by the simulated mid-Holocene expansion of dynamic lakes and dynamic vegetation (mHdVdL-mH: Fig. 7a) is small compared to the the total precipitation increase caused by all considered mid-Holocene forcings combined (mHdVdL-pidVdL: Fig. 6a).

A factor analysis shows that the precipitation increase from dynamic vegetation alone and dynamic lakes alone (Fig. 7b-c) is, in the sum, higher than the precipitation increase from dynamic lakes and vegetation together (Fig. 7a). Accordingly, the synergy effect of lakes and vegetation causes a drying response over the entire Sahel (Fig. 7d). Results show that the dynamic vegetation expansion alone causes a higher precipitation increase over the Sahel (Fig. 7b), and the dynamic lake expansion alone causes a higher precipitation increase to the south of Chad Lake (Fig. 7c) compared to the precipitation changes induced

by dynamic lakes and dynamic vegetation together (Fig. 7a). These results raise the question about what causes the synergy effect that leads to a drying response over the Sahel.





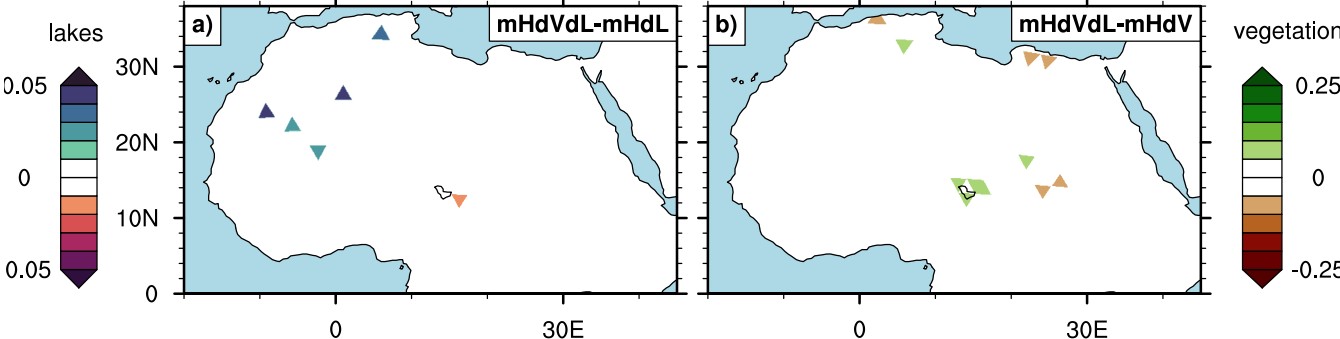

**Figure 8.** Simulated mid-Holocene (a) the lake extent changes due to the mid-Holocene vegetation cover extent (mHdVdL-mHdL) and (b) the vegetation cover changes due to the mid-Holocene lake extent (mHdVdL-mHdV).

Results indicate that the drying response from lake-vegetation synergies must come from other factors than the competition in the areal extent between lakes and vegetation. In fact, comparison between the individual mid-Holocene simulations reveals a weak, but predominately positive feedback between the areal extent of lakes and vegetation. For example, the simulated lake

extent enhances the local precipitation over Lake Chad (Fig. 7a-b), which results in a small expansion in vegetation in the vicinity of Lake Chad (Fig. 8b). Similarly, the simulated mid-Holocene vegetation extent enhances the rain fall in the western Sahara (Fig. 7a,c), which increases the lake extent in this region (Fig. 8a). Only the extent of Lake Chad slightly decreases (Fig. 8a), which is related to the rain fall decrease in the southern part of the Chad Catchment (Fig. 7a,c: south of Lake Chad). The differences in the lake and vegetation extent between the individual mid-Holocene simulations are overall small (Fig. 8)

compared to the overall mid-Holocene lake and vegetation extent changes (Fig. 6b-c: please note the different in the scaling). Thus, the predominately positive feedback between the lake and vegetation extent seems to have only a small influence on the mid-Holocene precipitation and, therefore, other factors must cause the drying response over the Sahel.

The drying response over the Sahel caused by the synergy effect of lakes and vegetation likely relates to changes in the meridional surface-temperature gradient resulting from a warming response over Lake Chad (Fig. 9c: solid line). The change

in the meridional surface temperature gradient lead to an overturning circulation response with near-surface meridional winds blowing from the colder land towards the warmer lake (Fig. 9b: arrows). The southerly wind response in the north of Lake Chad causes, by Coriolis force, an easterly wind acceleration at about 12 °N-16 °N (Fig. 9a: blue shading). Additionally, the convergence of meridonal winds above Lake Chad appears to be balanced not only by ascending motions (Fig. 9c: arrows), but also by an acceleration of the easterly winds between 8 °N and 16 °N (Fig. 9a: blue shading). The resulting total easterly

wind response over Lake Chad counteracts the near-surface westerly monsoon winds that transport moisture from the tropical Atlantic to the African continent. Accordingly, the easterly wind response decreases the moisture availability above Lake Chad (Fig. 9c: orange shades), which dampens the rain-generating deep convection in the overlying mid- to upper troposphere (Fig. 9c: arrows).







**Figure 9.** Simulated mid-Holocene circulation response (left panel: a-c) to the synergy effect of interacting dynamic lakes and vegetation ($\delta_{syn} = mHdVdL + mH - mHdV - mHdL$: see Eq. 1c), and (right panel: d-e) to a decreased lake depth, but unchanged area of Lake Chad ($\delta_{depth} = mH - mHL10$: see Eq. 2). Values are zonally averaged over 10 °N-20 °N for the boreal summer months (JJAS). The upper panel (a,d) shows the zonal wind response (colored shades) and the zonal wind climatology of the mH simulation (black contours). The middle panel (b,e) shows the vertical and meridional wind response (arrows) and specific humidity response (colored shades). Note that the vertical wind component was re-scaled (multiplied by 150) for visibility reasons. The lower panel (c,f) shows the surface temperature response (solid line). The dashed line in plot c shows the surface temperature changes caused by the mid-Holocene lake expansion in a green Sahara, which is given by $mHdVdL - mHdV$. The 'Lake Chad' labels indicate the location of Lake Chad, which is about 11 °N- 15 °N.

In addition, changes in the meridional surface-temperature gradient lead to a near-surface divergence to the south of Lake
Chad at about 5 °N (Fig. 9c: arrows). This near-surface divergence reduces the moist convection in the lower troposphere, which dampens the rain-generating deep convection aloft in the mid- to upper troposphere at about 5 °N and by that weakens the subsequent flow of the westerly monsoon winds (Fig. 9c: arrows). This reduction in the deep convection additionally





decreases the precipitation over tropical Africa. However, the main reason for the rainfall decrease is the Sahel is easterly wind acceleration above Lake Chad (Fig. 10a).

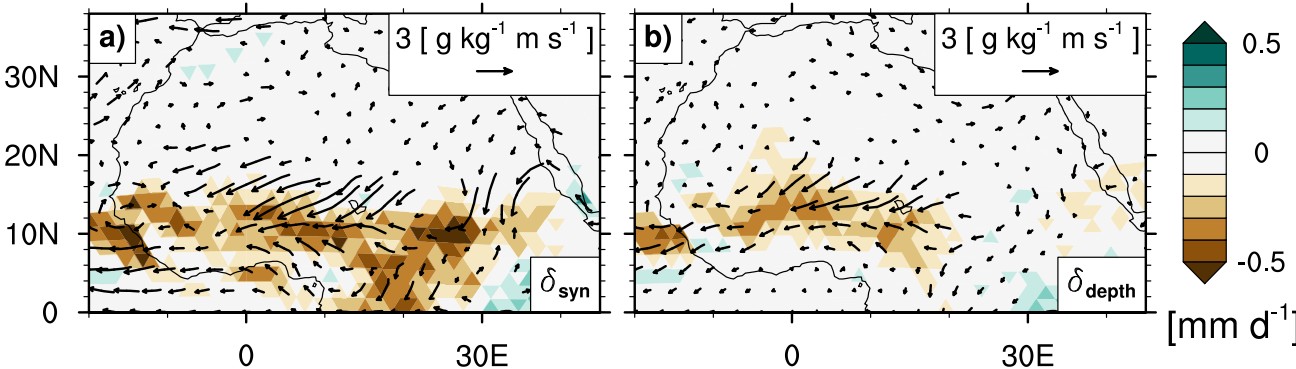

**Figure 10.** Simulated mid-Holocene moisture flux response (arrows) and precipitation response (colored shades) at 925 hPa to (a) the synergy effect to lakes and vegetation (mHdVdL + mH - mHdV - mHdL), and (b) a decrease in the depth of Lake Chad (mH - mHL10) during the summer months (JJAS).

Results show that the easterly wind acceleration weakens the monsoon westerly winds not only locally, but also affects the region upstream of Lake Chad to the west. Accordingly, the inland moisture flux decreases in these regions and rain-producing deep convection is reduced (Fig. 10a: arrows). The decrease in the near-surface inland moisture flux leads to a drying response in the entire Sahel along 12 °N (Fig. 10a: brown shades).

        Idealized mid-Holocene experiments show that a similar drying response over the Sahel is induced when the depth of Lake
Chad is decreased by about 1-5 m, without changing its spatial extent (Fig. 10b; Fig. 11b). By reducing the depth of Lake Chad from a standard model depth of 10 meter (as in former simulation studies: Specht et al., 2022) to the actually simulated, shallower model depth, the lake's heat storage capacity decreases, which leads to a faster warming of Lake Chad during the summer months (Fig. 9f). In contrast, the surface albedo of 0.07 and the extent of the evaporating surface remains unchanged. This indicates that the surface temperature of the lakes sensitively depends on the simulated lake depth and that the lake depth,
therefore, sensitively influences the meridional surface-temperature gradient and precipitation response at the latitude of the simulated lake.

        In the idealized mid-Holocene experiments, the changes in the meridional surface-temperature gradient induce an overturning circulation response similar to the one caused by the synergy effect of lakes and vegetation (Fig. 9b,e). This circulation response includes a near-surface easterly wind acceleration above the Lake Chad that decreased the rainfall around 12 °N and
a dipole-like zonal wind response in the overlying mid-troposphere corresponding to a southward shift of the African Easterly Jet (Fig. 9d-e) and, thus, a southward shift of the rain belt's northern boundary. Beside these similarities between both experiments, small differences exist regarding the response amplitude and the latitudinal position of the maximum easterly wind acceleration and the mid-troposphere dipole-like zonal wind response.




The differences in the latitudinal position of the atmospheric response occurs because a different extent of Lake Chad is
considered in the idealized lake-depth experiments and the dynamic interaction experiments. In the idealized lake-depth exper-
iments the extent of Lake Chad is kept fixed at its simulated pre-industrial value, and in the dynamic interaction experiments
the effect of the mid-Holocene lake expansion is included. In the latter experiments, the largest lake extent is found at the
southern boundary of the Lake Chad basin. Accordingly, the atmospheric response in the dynamic interactive experiments is
shifted somewhat southward compared to the idealized lake-depth experiments.

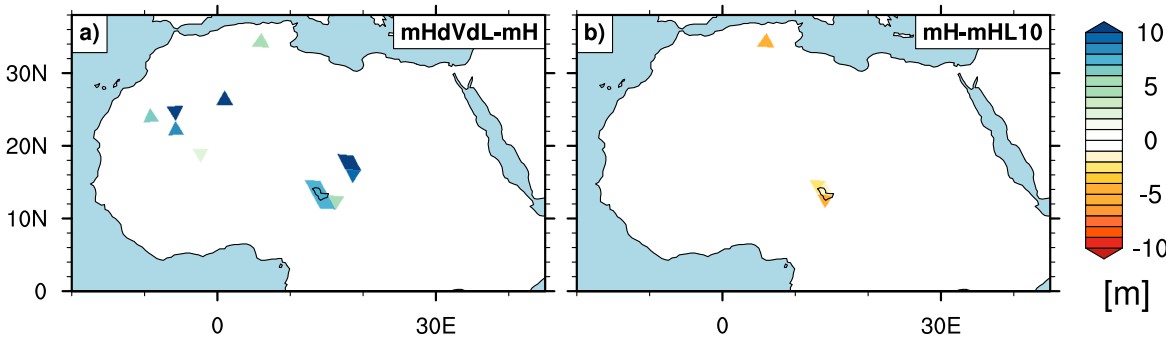

**Figure 11.** Differences in the lake depth (a) between the simulated mid-Holocene and pre-industrial lakes (mHdVdL - mH) and (b) between
the simulated pre-industrial lakes and the prescribed standard lakes with a constant 10-m lake depth (mH - mHL10).

The area of Lake Chad is much larger in the mid-Holocene simulation than in the pre-industrial simulation. But the simulated
mid-Holocene Lake Chad is rather shallow with a depth of about 5-8 m at its southern basin at around 12 °N (Fig. 11a). This
large but shallow Lake Chad has a relatively warm surface temperature compared to the vegetated land surface in this region.
Comparison between the simulations mHdVdL and mHdV shows that replacing the simulated mid-Holocene vegetation extent
with the simulated mid-Holocene lake extent in the region of Lake Chad results in a local surface warming, i.e. the simulated
mid-Holocene vegetation cools the land surface stronger than the simulated mid-Holocene Lake Chad (Fig. 9d: dashed line).
This warming effect caused by the shallow extent of Lake Chad contributes to large part to the total warming response caused
by the synergy effect of lakes and vegetation at around 12 °N (Fig. 9d: dashed vs. solid line).

Thus, in contrast to the simulated mid-Holocene extent of Lake Chad in a desert landscape that causes local surface cooling
and precipitation increase (Fig. 7c), the simulated mid-Holocene extent of Lake Chad in a vegetated northern Africa causes a
surface warming and, therefore, a drying response (Fig. 9c).

## 3 Conclusions

Our results show that the DEL model realistically simulates the pre-industrial extent of Lake Chad and the presence of some
dry basins over northern Africa. Only the extent of Lake Ahnet and Lake Chotts, which receive most of their discharge from
the Atlas Mountains, is overestimated by about $1,512\,\text{km}^2$ and $2,004\,\text{km}^2$, respectively. The overestimated lake extent of both



lakes likely is caused by the coarse resolution of the simulation. The coarse resolution limits the accurate representation of the catchment boundaries and the accurate representation of the small-scale heterogeneous precipitation from orographic updrafts, as occurs in the Atlas Mountains.

Additionally, the DEL model only mimics the effect of an aquifer reservoir to avoid unrealistic high inter-annual fluctuations in the lake depth and lake extent. In the HD model, though, a comprehensive aquifer model that interacts with the overlying

dynamic lakes is missing. Yet, reconstructions indicate a complex interaction between endorheic lakes and the underlying aquifer reservoirs (Lézine et al., 2011b). This interaction causes, for example, a 3,000-year time-lag between the orbital-forced summer insolation maximum and the maximum lake extent, because the aquifers over northern Africa filled during the early Holocene and recharged the overlying lakes during the mid-Holocene and towards the end of the African Humid Period (Lézine et al., 2011b). This interaction is not explicitly simulated by the DEL and HD model. Therefore, we assume that the DEL model

is likely not suitable for transitional simulations, like the simulation of the delayed lake decline over northern Africa at the End of the African Humid Period. But our results indicate that the DEL model realistically simulates the mean state of the endorheic lakes over northern Africa with regard to the terrestrial water balance.

The simulated mid-Holocene Lake Chad with an area of $67,480$ km$^2$ is strongly underestimated compared to the reconstructed mid-Holocene Lake Chad with an area of about $350,000$ km$^2$ (Hoelzmann et al., 1998; Quade et al., 2018; Drake

et al., 2022). Comparison between reconstructions and simulated lakes in the western Sahara is difficult because lake reconstructions from this region are subject to large uncertainties (Drake et al., 2022). Simulated lakes in the western Sahara are presumably also underestimated by the ICON-JSBACH4 model. The underestimated lake extent likely relates to the a known dry bias over northern Africa simulated by the ICON-ESM (Schneck et al., 2022). Climate models that produce a more humid mid-Holocene climate over northern Africa, like the MPI-ESM (e.g. Dallmeyer et al., 2021), potentially simulate a much larger

vegetation and lake extent over northern Africa. Even though the lake and vegetation extent in our mid-Holocene simulations is underestimated, the simulations provide interesting insights into what influences the interaction between climate, lakes and vegetation over northern Africa during the mid-Holocene.

The factor analysis indicates that the meridional surface-temperature gradient between the dynamic lakes and dynamic vegetation sensitively influences the mid-Holocene precipitation over northern Africa and, therefore, plays a major role for

the interaction between the West African summer monsoon, lakes and vegetation during the mid-Holocene. For example, the simulated mid-Holocene extent of vegetation in the Sahel cools the land surface stronger than the simulated mid-Holocene extent of Lake Chad. Accordingly, the simulated mid-Holocene extent of Lakes Chad in a densely vegetated Sahel leads to a local surface warming. The resulting local changes in the meridional surface-temperature gradient induce an overturning circulation response in the lower troposphere that decreases the inland moisture transport from the tropical Atlantic into the

African continent and leads to a drying response in the entire Sahel. This drying response contradict the results of former simulations studies, which show a general precipitation increase in response to a mid-Holocene extent of Lake Chad prescribed from reconstructions (Coe and Bonan, 1997; Broström et al., 1998; Carrington et al., 2001; Krinner et al., 2012; Chandan and Peltier, 2020; Specht et al., 2022; Li et al., 2023). This precipitation increase also occurs in the study by Specht et al. (2022), who even use the same climate model, ICON-JSBACH4. Specht et al. (2022), though, prescribe a constant lake depth of 10



meters, which is the standard lake depth of the ICON-JSBACH4 model. In our study the lake depth is treated dynamically and
depends on the local water budget.

Idealized mid-Holocene simulations show that the simulated lake depth sensitively influences the surface temperature of the
dynamic lakes and, thus, the local meridional surface-temperature gradient between the lakes and their vicinity. For example,
our results show that decreasing the depth of Lake Chad, without changing its spatial extent, induces a local warming response

over Lake Chad. By reducing the depth of Lake Chad by about 1-5 m, the heat storage capacity of the lake decreases, which
leads to a faster warming of the lake during the summer months. The resulting local surface warming causes a circulation and
drying response that is similar to the one caused by the mid-Holocene extent of Lake Chad in a densely vegetated Sahel. Based
on these results, we conclude that a proper representation of lake parameters - like the lake depth - is important to properly
simulate the surface temperature of lakes over northern Africa and to investigate the effect of lakes on the mid-Holocene

monsoon precipitation during the mid-Holocene.

Unfortunately, previous mid-Holocene simulation studies provide only little information on how lakes are represented in
terms of the lake depth or the lake surface albedo (Coe and Bonan, 1997; Broström et al., 1998; Carrington et al., 2001;
Krinner et al., 2012; Chandan and Peltier, 2020; Li et al., 2023). Since lake reconstructions rarely provide information about
the lake depth (e.g. Hoelzmann et al., 1998), the effect of the mid-Holocene lake depth changes on the monsoon precipitation

is likely neglected in previous simulation studies (Coe and Bonan, 1997; Broström et al., 1998; Carrington et al., 2001; Krinner
et al., 2012; Chandan and Peltier, 2020; Specht et al., 2022; Li et al., 2023). The differences in the lake representation between
different climate models might be the reason why in some models, a prescribed mid-Holocene lake extent causes a local and
marginal precipitation increase only (Coe and Bonan, 1997; Broström et al., 1998; Chandan and Peltier, 2020), whereas the
same lake extent causes a substantial precipitation increase across northern Africa in other models (Krinner et al., 2012; Specht

et al., 2022; Li et al., 2023). In our study, lakes are treated as a pure mixed layer that has a dynamic depth and a constant
surface albedo of 0.07. A more realistic lake surface temperature might be simulated by taking into account the existence of a
lake thermocline and a dynamic lake albedo.

Our results suggest that the mid-Holocene lake extent only causes a local positive vegetation feedback over northern Africa.
In our simulations, though, the extent and deepening of mid-Holocene lakes is underestimated compared to reconstructions

(Hoelzmann et al., 1998; Quade et al., 2018; Drake et al., 2022). Therefore, the surface cooling by the simulated mid-Holocene
lakes is likely underestimated, too. A larger lake extent and lake deepening over northern Africa might cause a positive vege-
tation feedback if these simulated lakes cool the land surface stronger than the simulated vegetation of the surrounding land.
Thus, rather than showing the most realistic mid-Holocene precipitation increase due to the simulated extent of dynamic lakes
and dynamic vegetation, our study provides insights into what influences the dynamic interaction between climate, lakes and

vegetation over norther Africa.

Additionally, in our study we neglect the existence of wetlands, as well as the effect exorheic lakes, like Mega-lake Timbuktu
in the Niger River Inland Delta (Drake et al., 2022). Wetlands might cause a relatively high evaporation and surface cooling,
because this land surface type combines the high surface roughness of the vegetation with the moisture saturated surface of
lakes (Specht et al., 2022). A strong surface cooling due to the extent of wetlands potentially occurred in the vicinity of Mega-



Lake Chad (Hoelzmann et al., 1998), and in the western Sahel and Sahara (Chen et al., 2021). To gain a more comprehensive understanding on the effect of dynamic wetlands, more research is needed.

*Code and data availability.* A summary of the ICON-A JSBACH4 model version, the scripts to generate the boundary conditions and the run scripts and output data to generate the figures shown in this study are available at https://hdl.handle.net/21.11116/0000-000E-400F-A (last access: 18 January 2024).

**Appendix A**

*Author contributions.* NFS and MC planned the study. NFS constructed and implemented the DEL model, ran the simulations and analyzed the results. MC and TK contributed to the discussion of results and the paper.

*Competing interests.* At least one of the (co-)authors is a member of the editorial board of Climate of the Past.


*Acknowledgements.* This study contributes to the Cluster of Excellence "Climate, Climate Change, and Society" (CLICCS). The model simulations were performed at the German Climate Computing Center (DKRZ). Many thanks go to Reiner Schnur (MPI-M), Veronika Gayler (MPI-M), Tobias Stacke (MPI-M) and Thomas Riddick (MPI-M) for their excellent technical support and discussion.

Thomas Kleinen is funded by the project PalMod of the German Federal Ministry of Education and Research (BMBF), Research for
Sustainability initiative FONA (grant no. 01LP1921A). Nora Farina Specht is financed by the International Max Planck Research School on Earth System Modeling (IMPRS-ESM), Hamburg.



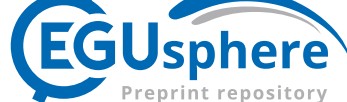

**Figure A1.** Time series of simulated mid-Holocene (blue) lake and (green) vegetation extent for the (a) mHdVdL, (b) mHdL, (c) mHdV, and (d) mHdVdLmax simulation. The blue line shows the lake area within the endorheic catchments of North Africa and the green line shows the vegetation cover averages over the Sahel and Sahara (20 °E-35 °W,10 °N-35 °N). The time series section with the white background part shows the evaluation period used for the analysis. Depending on the variability of the lake area and vegetation cover, the evaluation period is 150 or 200 year long.



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
