# Peer review of "Dynamic interaction of lakes, climate and vegetation over northern Africa during the mid-Holocene"

_EGUsphere, 2023_

## Referee Comment (RC2)

**Manuscript:** Dynamic interaction of lakes, climate and vegetation over northern Africa during the mid-Holocene

**Major remarks**

The authors investigated the effect of large lakes in the Sahel and Sahara on the West African summer monsoon during the mid-Holocene. In order to ensure consistency between the lake expansion and the simulated climate and the associated water balance, they developed a dynamic endorheic lake (DEL) model and implemented it into the atmosphere-land model ICON-JSBACH4. Their mid-Holocene simulations showed that both, lake and vegetation expansion during the mid-Holocene caused a precipitation increase over northern Africa, while the lake-vegetation interaction is somewhat counteracting the overall effect with a relative drying response over the entire Sahel. The study is well written and contributes valuable insights into mid-Holocene dynamics and land-atmosphere interactions over northern Africa.

I have only one major remark that is related to Fig. 11. It shows only relatively small areas where the lake depth is changed. However, the results show a comparatively large effect. Can't it be that the effect is related to natural variability? In order to consider this, you may separate your 150-200 years evaluation period into 30-year time slices and check whether the impact of the lake depth change is a robust feature or has a larger natural variability.

I suggest adding a table with regional averages (e.g. Sahel and Sahara) in precipitation for the different experiments that would allow an easy comparison of the different precipitation changes (see also my comment to Line 323-325).

I also suggest a thorough proof reading as the current version of the manuscript comprises several typos. Below I suggested correcting those I found.

In summary, I suggest accepting the paper for publication after minor revisions are conducted.

**Minor remarks**

In the following suggestions for editorial corrections are marked in *Italic*.

Line 11
… Basin*, the lake area is* slightly …

Line 16
… lake expansion *that is dominated by* the expansion …

Line 16-18
Sentence is somewhat difficult to read. Please rephrase, e.g. separate into two sentences!

Line 54
… treated, *all previous simulation studies* prescribe …

This statement sems to be too general. I guess you mean studies with GCMs? With regard to hydrology, there are several studies that use climate forcing to simulate lake area expansion in the mid-Holocene, e.g. Coe 1997: Stacke 2011.

Coe, M.: Simulating continental surface waters: An application to holocene Northern Africa, J. Climate, 10, 1680–1689, 1997.

Stacke, T. (2011). Development of a dynamical wetlands hydrology scheme and its application under different climate conditions. Phd Thesis, Hamburg: University of Hamburg. Berichte zur Erdsystemforschung, 99

Line 69
In the end of sect. 1, an outline about the following sections is missing. The last sentence only describes the content of Sect. 2.

Line 91
…applied *over northern* Africa.

Line 142
… concept of *the* endorheic …

Line 214
… *represented as* a mixed …

Line 228
…*orography is* used …

Line 242
… less *sensitive* to …

Line 245
… growth and *shrinking* of …

p.12 – Figure 5 caption – last sentence
It is written:
"The black boundary in subplot c) …"

I assume you mean panel d) not c)?
In addition, I suggest writing '*panel'* instead of 'subplot' throughout the paper.

Line 272
Considering *these* large …

Line 313
… presence *of dynamic lakes in the* ICON-JSBACH4 …

Line 324
… *to the* total …

Line 323-325
I suggest providing some values (e.g. averaged over the Sahel) to allow an easy comparison of the precipitation changes.

p. 16 – Fig. 8 caption:

*Simulated a) lake* extent changes …

Line 333
In fact, *the* comparison …

Line 340
… the *different scaling*).

Line 355 and 357
There are no arrows in Fig. 9c. Please correct!

Line 373-376
Sentence is too long and difficult to read. Please rephrase into two sentences.

Line 385
*However,* the simulated …

Line 391
…contributes *a* large …

Line 402
… updrafts, *which* occurs …

Line 406-408
Sentence is difficult to read and understand. Please rephrase!

Line 417
…relates to *the known* dry …

Line 447-457
This paragraph comprises the same or similar sequence of references several times. Please rephrase and avoid redundant use of the same references if possible.

Line 465
… over *northern* Africa.

Line 466
… effect *of* exorheic …

---

## Author Comment (AC1)

**Manuscript:** Dynamic interaction of lakes, climate and vegetation over northern Africa during the mid-Holocene

**Response to Reviewer 1**

*This is a well-written manuscript addressing a question which is certainly relevant to the scope of CP. It presents a new dynamic lake model implemented to address a pertinent question – the effect of lakes on the mid-Holocene northern African climate. To the best of my knowledge, this is a novel contribution, and it does lead to interesting new insights regarding the synergistic effect between vegetation and lakes as well as regarding the effect of lake depth. I do not have the required expertise to assess the HD and DEL models deeply; on the surface they appear to be reasonably formulated. Overall, I found the manuscript thoughtfully written and would recommend it for publication after addressing some issues.*

**We would like to thank the reviewer for reading the manuscript carefully and for writing this valuable and constructive feedback. We are pleased to hear that the reviewer thinks the manuscript is thoughfully written and recommends it for publication. In the following, we address the reviewer's comments point by point. We respond with "Ok." in case we would correct the manuscript exactly as suggested.**

*Specific comments:*

*Line 265: I'm not sure if the lake and vegetation cover can be claimed to have reached a "close-to-equilibrium state" at 400 years into the simulation. Looking at Figure 5, the choices of last 100 years or the last 400 years as evaluation period seem more logical to me, compared to last 150 years which starts with a declining peak in both covers. The peak in lake cover at the beginning of the evaluation period is around 10,000 km$^2$ and that doesn't seem trivial to me. Either the authors could explain the quantitative basis behind this choice of evaluation period, or remove the phrase "close-to-equilibrium state" and simply mention that the last 150 years were used as evaluation period. Does the precipitation over northern Africa change significantly if the last 400 years are used instead?*

**We will  remove the phrase "close-to-equilibrium state" and simply mention that the last 150 years were used as evaluation period. But we will check whether it makes a large difference when using the last 400 years instead of 150 years.**

*Figure 5: Please mention in the caption what the horizontal dashed lines in subplot a) represent.*

**We will add the information that the blue and green dashed line ind figure 5 a) represents the mean lake area and mean vegetation cover averaged over the last 150 years.**

*Lines 270 – 273: I appreciate this example to put the 18.8% deviation in context.*

**Thank you!**

*Line 303: Some newer work has indicated that precipitation could be even higher than that indicated by Bartlein et al. (2011) (which is also the proxy dataset that Braconnot et al., 2012 refer to). For example, Hely et al. (2014) show that Sudanian taxa were present up to 25 N, which would require 500-1500 mm/yr of rainfall, and Guineo-Congolian taxa (requiring >1500 mm/yr) potentially reached 20 N. More "extreme" examples are also provided by Tierney et al. (2017) and Sha et al. (2019). Hence, I suggest changing "seems to underestimate" to "underestimates" since that is a clear and significant under-estimation of precipitation.*

**We will add the suggested citations to the results part and changes the term "seems to underestimate" to "underestimates".**

*Line 312: Braconnot et al. (2012) discussed PMIP1 and PMIP2 model results. Brierley et al. (2020) might be a better reference for state-of-the-art climate models. However, in my understanding, biases are usually discussed in terms of simulation of present day climate, and not while discussing (MH-PI, or similar) anomalies.*

**Thank you for the suggestion. We will include Brierly et al. (2020) in our text.**

*Line 328: Could the authors expand a bit on why lake expansion leads to higher precipitation south of Megalake Chad?*

**We will add the following information at line 325: "The dynamic vegetation alone causes a local precipitation increase in regions where the vegetation expands over northern Africa (Fig. 6c; Fig. 7b). This precipitation increase is likely caused by an enhanced local moisture recycling (citation). In contrast, Lake Chad not only enhances precipitation locally above its water surface, but also to its southern side around 5 °N (Fig. 6b; Fig. 7c). This precipitation increase is associated with winds that blow from the cool lake towards its warmer surrounding, where it converges with the monsoon westerly wind to the south of Lake Chad. The resulting moisture convergences to the south of Lake Chad enhances moist convetion and, thus, the formation of precipitation in this region, which again rehances the inflow of monsoon westerly winds. The related mechanism is described later in the results part."**

Line 390: I think a plot similar to Figure 7 but for mean annual temperature would help visualize this point. Additionally, could the authors provide estimates for changes in albedo due to vegetation and lakes? What exactly causes the change in the meridional temperature gradient?

**We will compute a factor analysis for temperature and albedo changes and see if it helps to visualize that „the simulated mid-Holocene vegetation cools the land surface stronger than the simulated mid-Holocene Lake Chad". We will add this plot to the appendix if applicable. Another, more simple, approach would be to add the meridional mean changes in the albedo to plot 9 at the bottom, where the suface temperature changes of mHdV dL − mHdV are already displayed.**

**We will also provide the information that the lake albedo is set constant to 0.07 and the range in the albedo for the vegetation in the model.**

**The surface temperature and, thus, its meridional gradient are influced mainly by the albedo (net radiation) and turbulent heat fluxes (latent and sensible heat flux).**

*Line 430: I'm not sure the results of this study strictly "contradict" the previous studies, since none of the previous studies discussed the synergistic effect, but only the net effect of vegetation and lakes. If I've understood correctly, the drying response is only due to the synergistic effect, and this study agrees with previous ones regarding the net effect.*

**We agree on this comment and will rephrase this sentence as the following: " This drying response is an unexpected result, since former simulation studies show a general precipitation increase in response to a mid-Holocene extent of Lake Chad prescribed from reconstructions (Coe and Bonan, 1997; Broström et al., 1998; Carrington et al., 2001; Krinner et al., 2012; Chandan and Peltier, 2020; Specht et al., 2022; Li et al., 2023)."**

Line 456: It seems to me that one of the main conclusions of the study (overturning circulation over Megalake Chad, and associated changes) is sensitive to the choice of the lake surface albedo. How much would this change with the use of a dynamic lake albedo? For example, what could be a plausible albedo range for Megalake Chad if a dynamic scheme were to be used?

**Thank you very much for this suggestion. We will add some information from observation estimates about the possible albedo range of lakes i.e. "The values of lake albedo in the study area ranged from 0 to 0.45, with an average value of 0.14." (Du et al. 2023)**

**Du, Jia, et al. "Retrieval of lake water surface albedo from Sentinel-2 remote sensing imagery." *Journal of Hydrology* 617 (2023): 128904.**

*Line 460-463: The switch from surface warming to surface cooling due to lakes is confusing.*

**We are not sure, if we understand this comment right. We would suggest to rephrase the sentence starting at line 460 as the following: „Since the depth of the lakes is likely underestimated, the surface cooling by the simulated mid-Holocene lakes is likely underestimated, too."**

*Technical comments:*

*The authors should consider merging Figures 2 and 3 into one figure with the differences highlighted. Results should be moved into a separate section (Section 2.4 ◊ Section 3), moving Conclusions forward (Section 3 ◊ Section 4). The authors could consider moving Figure 11 into the Supplementary Information.*

**We will show Figures 2 and 3 side by side and highlight the differences. Something must have gone wrong in the LaTeX template we used. we will number the individual chapters as follows: 1. introduction, 2. methods, 3. results and 4. conclusions. We will put Figure 11 – as suggested - in the appendix, as this figure is only briefly mentioned in the results.**

*Lastly, I found some grammatical/typographical errors which should be corrected:*

*Line 21: warms → warm*

**Ok.**

Line 67: In the following *section*?

**Ok.**

*Lines 79-81: Consider replacing the phrase "a too" with "overly" or "unrealistically"*

**We will replace "a too" with "an unrealistically"**

*Line 89: albedo scheme represents changes*

**Ok.**

*Line 91: over the northern Africa → over northern Africa*

**Ok.**

Line 110: on the *climate variables* e.g. the precipitation?

**Ok.**

*Line 142: thr → the*

**Ok.**

*Figure 1 caption: In the last sentence – endorheic catchments were generated in a way that (remove "it") suits*

**Ok.**

Line 214: represented by → represented as a

**Ok.**

*Line 245: shirk → shrink?*

**Ok.**

*Line 263: different → difference*

**Ok.**

*Figure 5 caption: In the first sentence – norther**n**. Last sentence – subplot c) or d)?*

**In the first sentence, we will change "norther" to "northern" and in the last senstence we will correct "c)" to "d)".**

*Line 264: 550 year**s***

**Ok.**

*Line 273: Considering this → such large changes?*

**We will change "Considering this" to "Considering such large changes"**

*Line 290: orbit → orbital*

**Ok.**

*Lines 334, 341: predominately → predominantly*

**Ok.**

*Line 340: note the different → difference in the scaling*

**Ok.**

*Lines 345, 348: meridonal → meridional*

**Ok.**

*Lines 348 onwards: please check all references to various subplots of Figure 9*

**There appears to be a systematic error in the reference to Firgure 9 b), which is incorrectly referenced as 9 c). We apologize for the mistake. We will correct the following:**

- **Line 348 "(Fig. 9c: arrows)" to "(Fig. 9b: arrows)"**
- **Line 352 "(Fig. 9c: orange shades)" to "(Fig. 9b: orange shades)"**
- **Line 353 "(Fig. 9c: arrows)" to "(Fig. 9b: arrows)"**
- **Line 255 "(Fig. 9c: arrows)" to "(Fig. 9b: arrows)"**

- **Line 257 "(Fig. 9c: arrows)" to "(Fig. 9b: arrows)"**

*Line 358: is → in the Sahel*

**We will change "is" to "in"**

*Line 410: transitional → transient?*

**Ok.**

Line 424: for → in

**Ok.**

Line 466: the effect *of* exorheic lakes

**Ok.**

*I appreciate the work put in by the authors and hope they find these comments helpful. I thank Prof. Buizert for considering me for this review.*

**Thank you again for reviewing this manuscript.**

---

## Author Comment (AC2)

**Manuscript:** Dynamic interaction of lakes, climate and vegetation over northern Africa during the mid-Holocene

**Response to Reviewer 2**

**Major remarks**

*The authors investigated the effect of large lakes in the Sahel and Sahara on the West African summer monsoon during the mid-Holocene. In order to ensure consistency between the lake expansion and the simulated climate and the associated water balance, they developed a dynamic endorheic lake (DEL) model and implemented it into the atmosphere-land model ICON-JSBACH4. Their mid-Holocene simulations showed that both, lake and vegetation expansion during the mid-Holocene caused a precipitation increase over northern Africa, while the lake-vegetation interaction is somewhat counteracting the overall effect with a relative drying response over the entire Sahel. The study is well written and contributes valuable insights into mid-Holocene dynamics and land-atmosphere interactions over northern Africa.*

**We would like to thank the reviewer for taking the time to read through our manuscript and for giving us such constructive feedback. Below we will respond to the author's comments point by point. We respond with "Ok." if we would correct the manuscript exactly as suggested.**

*I have only one major remark that is related to Fig. 11. It shows only relatively small areas where the lake depth is changed. However, the results show a comparatively large effect. Can't it be that the effect is related to natural variability? In order to consider this, you may separate your 150-200 years evaluation period into 30-year time slices and check whether the impact of the lake depth change is a robust feature or has a larger natural variability.*

**Thanks a lot for your suggestion. Regarding the idealized lake-depth experiment (mH-mHL10) as show in Fig. 11 b) the lake depth, as well as vegetation and lake extent is prescribed and therefore does not change throughout the simulation. We expect that natural variability should not play a role for these results.**

**However, we will compute the standard deviations of lake depth and precipitation to get an idea whether or not the changes seen in Fig. 11 a) are significant and induce significant changes in precipitation.**

*I suggest adding a table with regional averages (e.g. Sahel and Sahara) in precipitation for the different experiments that would allow an easy comparison of the different precipitation changes (see also my comment to Line 323-325).*

**We will provide the regional precipitation averages for the individual basins over northern Africa in an extra table in the appendix.**

*I also suggest a thorough proof reading as the current version of the manuscript comprises several typos. Below I suggested correcting those I found.*

**We will correct the typos as suggested.**

*In summary, I suggest accepting the paper for publication after minor revisions are conducted.*

**Minor remarks**

In the following suggestions for editorial corrections are marked in Italic.

*Line 11: … Basin, the lake area is slightly …*

**Ok.**

*Line 16: … lake expansion that is dominated by the expansion …*

**Ok.**

*Line 16-18: Sentence is somewhat difficult to read. Please rephrase, e.g. separate into two sentences!*

**We will rephrase this sentence as the follwoing: "Accordingly, the surface temperature increases over the region of Lake Chad and causes local changes in meridional surface-temperature gradient. These changes in the meridional surface-temperature gradient are associated with a reduced inland moisture transport from the tropical Atlantic into the Sahel, which causes a drying response in the Sahel."**

*Line 54: … treated, all previous simulation studies prescribe …*

*This statement sems to be too general. I guess you mean studies with GCMs? With regard to hydrology, there are several studies that use climate forcing to simulate lake area expansion in the mid-Holocene, e.g. Coe 1997: Stacke 2011.*

*Coe, M.: Simulating continental surface waters: An application to holocene Northern Africa, J. Climate, 10, 1680–1689, 1997.*

*Stacke, T. (2011). Development of a dynamical wetlands hydrology scheme and its application under different climate conditions. Phd Thesis, Hamburg: University of Hamburg. Berichte zur Erdsystemforschung, 99.*

**Yes, this statement seems to be in fact too general or even missleading and therefore needs to be rephrased. We here wanted to refer to studies that investigated the effect of North African lakes on the mid-Holocene climate using GCM only. We would therefore rephrase the sentence as the following: "Apart from the difference in how the vegetation is treated, previous simulation studies all prescribe the mid-Holocene lake extent over the Sahel and Sahara from reconstructions to investigate there effect on the mid-Holocene climate (Li et al., 2023; Specht et al., 2022; Chandan and Peltier, 2020; Krinner et al., 2012; Broström et al., 1998; Coe and Bonan, 1997)."**

**We think the suggested citations (Coe 1997; Stacke 2011) would rather contribute to the statement at line 38-41, since these studies focus on modelling the mid-Holocene surface water extent using a prescribed mid-Holocene climate/hydrological forcing.**

*Line 69: In the end of sect. 1, an outline about the following sections is missing. The last sentence only describes the content of Sect. 2.*

**We are sorry for the missing outline. We will correct the numbering of the individual sections as follows: 1. introduction, 2. methods, 3. results and 4. conclusions. At the end of the introduction, we will give a brief overview of the following section: This could read as follows:**

**"In the following methods section, the concept of the dynamic lake model and the structure of the present-day and mid-Holocene simulations is described. In the results section, the simulated present-day lake extent is briefly evaluated by comparing it with observational data. Similarly, we compare the simulated mid-Holocene precipitation increase as well as the lake and vegetation extent with mid-Holocene reconstruction data. Additinally, we look at the individual and synergistic effect of the mid-Holocene lake and vegetation extent on the North African climate and how the depth of the lakes influences the mid-Holocene climate over North Africa. Finally, we discuss our results in relation to former studies in the conclusions part."**

*Line 91: …applied over northern Africa.*

**Ok.**

*Line 142: … concept of the endorheic …*

**Ok.**

*Line 214: … represented as a mixed …*

**Ok.**

*Line 228: …orography is used …*

**Ok.**

*Line 242: … less sensitive to …*

**Ok.**

*Line 245: … growth and shrinking of …*

**Ok.**

*p.12 – Figure 5 caption – last sentence: It is written: "The black boundary in subplot c) …" I assume you mean panel d) not c)? In addition, I suggest writing 'panel' instead of 'subplot' throughout the paper.*

**Yes, we mean panel d), not c). Sorry for the confusion. We will replace "subplot" with "panel" throughout the document as suggested.**

*Line 272: Considering these large …*

**Ok.**

*Line 313: … presence of dynamic lakes in the ICON-JSBACH4 …*

**Ok.**

*Line 324: … to the total …*

**Ok.**

*Line 323-325: I suggest providing some values (e.g. averaged over the Sahel) to allow an easy comparison of the precipitation changes.*

**We will provide these information in an extra table or as part of figure A1 in the appendix.**

*p. 16 – Fig. 8 caption: Simulated a) lake extent changes …*

**Ok.**

*Line 333: In fact, the comparison …*

**Ok.**

*Line 340: … the different scaling).*

**Ok.**

*Line 355 and 357: There are no arrows in Fig. 9c. Please correct!*

**We realized that Fig. 9b) is systematically refered to as 9c) by mistake. We will correct this error.**

*Line 373-376: Sentence is too long and difficult to read. Please rephrase into two sentences.*

**We agree that this sentence is hard to undertsnad. Therefore, we will rephrase it as the following: „This circulation response includes a near-surface easterly wind acceleration above Lake Chad that decreaseds the inland moisture transport (Fig. 9d-e) and, thus, rainfall at around 12 ◦N. Additionally, the circulation response includes a dipole-like zonal wind response in the mid-troposphere above Lake Chad that corresponds to a southward shift of the African Easterly Jet (Fig. 9d-e) and, thus, a southward shift of the rain belt's northern boundary."**

*Line 385: However, the simulated …*

**Ok.**

*Line 391: …contributes a large …*

**Ok.**

*Line 402: … updrafts, which occurs …*

**Ok.**

*Line 406-408: Sentence is difficult to read and understand. Please rephrase!*

**We will rephrase the sentence accordingly: "This interaction caused a 3,000-year time-lag between the orbital-forced summer insolation maximum and the maximum lake extent during the mid-Holocene (Lézine et al., 2011b). This time lag is due to the fact that the aquifers over North Africa filled up in the early Holocene until the water table reached the overlying lake basins, leading to the formation of larger lakes. As precipitation over northern Africa decreased towards the end of the African Humid Period, the lake basins continued to be fed by the aquifers and, therefore, regressed with a delay to the orbitally-forced precipiation changes."**

*Line 417: …relates to the known dry …*

**Ok.**

Line 447-457: This paragraph comprises the same or similar sequence of references several times. Please rephrase and avoid redundant use of the same references if possible.

*Line 465: … over northern Africa.*

**Ok.**

*Line 466: … effect of exorheic …*

**Ok.**

---

## Author Response (AR2)

**Manuscript:** Dynamic interaction of lakes, climate and vegetation over northern Africa during the mid-Holocene

**Response to Reviewer 1**

*This is a well-written manuscript addressing a question which is certainly relevant to the scope of CP. It presents a new dynamic lake model implemented to address a pertinent question – the effect of lakes on the mid-Holocene northern African climate. To the best of my knowledge, this is a novel contribution, and it does lead to interesting new insights regarding the synergistic effect between vegetation and lakes as well as regarding the effect of lake depth. I do not have the required expertise to assess the HD and DEL models deeply; on the surface they appear to be reasonably formulated. Overall, I found the manuscript thoughtfully written and would recommend it for publication after addressing some issues.*

**We would like to thank the reviewer for reading the manuscript carefully and for writing this valuable and constructive feedback. In the following, we address the reviewer's comments point by point. We respond with "Ok.", in case we would correct the manuscript exactly as suggested.**

*Specific comments:*

*Line 265: I'm not sure if the lake and vegetation cover can be claimed to have reached a "close-to-equilibrium state" at 400 years into the simulation. Looking at Figure 5, the choices of last 100 years or the last 400 years as evaluation period seem more logical to me, compared to last 150 years which starts with a declining peak in both covers. The peak in lake cover at the beginning of the evaluation period is around 10,000 $km^2$ and that doesn't seem trivial to me. Either the authors could explain the quantitative basis behind this choice of evaluation period, or remove the phrase "close-to-equilibrium state" and simply mention that the last 150 years were used as evaluation period. Does the precipitation over northern Africa change significantly if the last 400 years are used instead?*

**We have deleted the phrase "close-to-equilibrium state" and only mention that the last 150 years are used as evaluation period to obtain robust results. Mean annual precipitation and lake extent are slightly higher for the 400-year evaluation period than for the 150-year evaluation period. Annual precipitation in the Sahel is 15.9 mm per year$^{-1}$ higher for the 400-year evaluation period than for the 150-year evaluation period. Accordingly, the extent of Lake Chad is only marginally larger (2.7 %) for the 400-year evaluation period than for the 150-year evaluation period.**

*Figure 5: Please mention in the caption what the horizontal dashed lines in subplot a) represent.*

**We added the information that the blue and green dashed line in figure 5 a) represent the mean lake area and mean vegetation cover averaged over the last 150 years.**

*Lines 270 – 273: I appreciate this example to put the 18.8% deviation in context.*

**Thank you!**

*Line 303: Some newer work has indicated that precipitation could be even higher than that indicated by Bartlein et al. (2011) (which is also the proxy dataset that Braconnot et al., 2012 refer to). For example, Hely et al. (2014) show that Sudanian taxa were present up to 25 N, which would require 500-1500 mm/yr of rainfall, and Guineo-Congolian taxa (requiring >1500 mm/yr)*

*potentially reached 20 N. More "extreme" examples are also provided by Tierney et al. (2017) and Sha et al. (2019). Hence, I suggest changing "seems to underestimate" to "underestimates" since that is a clear and significant under-estimation of precipitation.*

**We have included the suggested citations and the associated precipitation changes in the results section and changed the term "seems to underestimate" to "underestimates".**

*Line 312: Braconnot et al. (2012) discussed PMIP1 and PMIP2 model results. Brierley et al. (2020) might be a better reference for state-of-the-art climate models. However, in my understanding, biases are usually discussed in terms of simulation of present day climate, and not while discussing (MH-PI, or similar) anomalies.*

**Thank you for the suggestion. We included Brierly et al. (2020) in our text.**

*Line 328: Could the authors expand a bit on why lake expansion leads to higher precipitation south of Megalake Chad?*

**We added the following information: "The dynamic vegetation alone causes a local precipitation increase in regions where the vegetation expands over northern Africa (Fig. 5c; Fig. 6b). This precipitation increase is caused by an enhanced local moisture recycling and a decrease in the surface albedo (e.g. Rachmayani et al., 2015). In contrast, Megalake Chad not only enhances precipitation locally above its water surface, but also in the region south of the megalake, around 5 ∘N (Fig. 5b; Fig. 6c). This precipitation increase is associated with winds that blow from the cool lake towards its warmer surrounding. To the south of Megalake Chad, the moist lake winds converge with moist westerly monsoon wind. Because of the moisture convergence to the south of Megalake Chad, moist convection and convective precipitation are enhanced. This in turn leads to an increase in the westerly monsoon wind."**

Line 390: I think a plot similar to Figure 7 but for mean annual temperature would help visualize this point. Additionally, could the authors provide estimates for changes in albedo due to vegetation and lakes? What exactly causes the change in the meridional temperature gradient?

**The added the figure A3 to the appendix to visualize that „the simulated mid-Holocene vegetation cools the land surface stronger than the simulated mid-Holocene Lake Chad". Figure A4 shows the differences in the mean summer surface temperature and mean summer albedo between the simulations mHdVdL and mHdV. In the results part we explain that the warming response is particularly strong, where the extent of Megalake Chad is shallowest and that the expansion of Lake Chad reduces the local surface albedo by about 0.02 to 0.04 compared to the vegetated land, which increases surface energy absorption and, thus, contributes to the warming response in the southern basin of Megalake Chad.**

*Line 430: I'm not sure the results of this study strictly "contradict" the previous studies, since none of the previous studies discussed the synergistic effect, but only the net effect of vegetation and lakes. If I've understood correctly, the drying response is only due to the synergistic effect, and this study agrees with previous ones regarding the net effect.*

**We agree on this comment and rephrased the sentence as the following: "This drying response is an unexpected result, since former simulation studies show a general precipitation increase in response to a mid-Holocene extent of Lake Chad prescribed from reconstructions (Coe and Bonan, 1997; Brostrom et al., 1998; Carrington et al., 2001; Krinner et al., 2012; Chandan and Peltier, 2020; Specht et al., 2022; Li et al., 2023)."**

Line 456: It seems to me that one of the main conclusions of the study (overturning circulation over Megalake Chad, and associated changes) is sensitive to the choice of the lake surface albedo. How much would this change with the use of a dynamic lake albedo? For example, what could be a plausible albedo range for Megalake Chad if a dynamic scheme were to be used?

**Thank you very much for this suggestion. We added the following information to the manuscript: "the albedo of lakes with a depth of >3 m might vary from 0.05 to 0.1 (Cogley, 1979), whereas the albedo of smaller water bodies with a depth of 1-3 m might vary from 0.09 to 0.22 (de Fleury et al., 2023) within the Sahelian-Saharan region (12◦N – 35◦N)."**

*Line 460-463: The switch from surface warming to surface cooling due to lakes is confusing.*

**We are not sure, if we understand this comment right. We rephrased the sentence starting at line 460 as the following: „Since the depth of the lakes is likely underestimated, the surface cooling by the simulated mid-Holocene lakes is likely underestimated, too."**

*Technical comments:*

*The authors should consider merging Figures 2 and 3 into one figure with the differences highlighted. Results should be moved into a separate section (Section 2.4 ◊ Section 3), moving Conclusions forward (Section 3 ◊ Section 4). The authors could consider moving Figure 11 into the Supplementary Information.*

**We decided to plot figure 2 and 3 side by side and highlighted the differences. In addition, we have corrected the numbers of the individual chapters as follows: 1. introduction, 2. methods, 3. results and 4. conclusions. We put Figure 11 – as suggested - in the appendix (A2), as this figure is only briefly mentioned in the results.**

*Lastly, I found some grammatical/typographical errors which should be corrected:*

*Line 21: warms→warm*

**Ok.**

Line 67: In the following *section*?

**Ok.**

*Lines 79-81: Consider replacing the phrase "a too" with "overly" or "unrealistically"*

**We replaced "a too" with "an unrealistically"**

*Line 89: albedo scheme represents changes*

**Ok.**

*Line 91: over the northern Africa → over northern Africa*

**Ok.**

Line 110: on the *climate variables* e.g. the precipitation?

**Ok.**

*Line 142: thr → the*

**Ok.**

*Figure 1 caption: In the last sentence – endorheic catchments were generated in a way that (remove "it") suits*

**Ok.**

Line 214: represented by → represented as a

**Ok.**

*Line 245: shirk → shrink?*

**Ok.**

*Line 263: different → difference*

**Ok.**

*Figure 5 caption: In the first sentence – norther**n**. Last sentence – subplot c) or d)?*

**In the first sentence, we changed "norther" to "northern" and in the last sentence we will corrected "c)" to "d)".**

*Line 264: 550 year**s***

**Ok.**

*Line 273: Considering this → such large changes?*

**We changed "Considering this" to "Considering these"**

*Line 290: orbit → orbital*

**Ok.**

*Lines 334, 341: predominately → predominantly*

**Ok.**

*Line 340: note the different → difference in the scaling*

**Ok.**

*Lines 345, 348: meridonal → meridional*

**Ok.**

*Lines 348 onwards: please check all references to various subplots of Figure 9*

**There was a systematic error in the reference to Figure 9 b), which was incorrectly referenced as 9 c). We apologize for the mistake. We corrected the following:**

- **Line 348 "(Fig. 9c: arrows)" to "(Fig. 9b: arrows)"**
- **Line 352 "(Fig. 9c: orange shades)" to "(Fig. 9b: orange shades)"**
- **Line 353 "(Fig. 9c: arrows)" to "(Fig. 9b: arrows)"**
- **Line 355 "(Fig. 9c: arrows)" to "(Fig. 9b: arrows)"**
- **Line 357 "(Fig. 9c: arrows)" to "(Fig. 9b: arrows)"**

*Line 358: is → in the Sahel*

**We changed "is" to "in"**

*Line 410: transitional → transient?*

**Ok.**

Line 424: for → in

**Ok.**

Line 466: the effect *of* exorheic lakes

**Ok.**

*I appreciate the work put in by the authors and hope they find these comments helpful. I thank Prof. Buizert for considering me for this review.*

**Thank you again for reviewing this manuscript.**

**Manuscript:** Dynamic interaction of lakes, climate and vegetation over northern Africa during the mid-Holocene

**Response to Reviewer 2**

**Major remarks**

*The authors investigated the effect of large lakes in the Sahel and Sahara on the West African summer monsoon during the mid-Holocene. In order to ensure consistency between the lake expansion and the simulated climate and the associated water balance, they developed a dynamic endorheic lake (DEL) model and implemented it into the atmosphere-land model ICON-JSBACH4. Their mid-Holocene simulations showed that both, lake and vegetation expansion during the mid-Holocene caused a precipitation increase over northern Africa, while the lake-vegetation interaction is somewhat counteracting the overall effect with a relative drying response over the entire Sahel. The study is well written and contributes valuable insights into mid-Holocene dynamics and land-atmosphere interactions over northern Africa.*

**We would like to thank the reviewer for taking the time to read through our manuscript and for giving us such constructive feedback. Below we will respond to the author's comments point by point. We respond with "Ok." if we would correct the manuscript exactly as suggested.**

*I have only one major remark that is related to Fig. 11. It shows only relatively small areas where the lake depth is changed. However, the results show a comparatively large effect. Can't it be that the effect is related to natural variability? In order to consider this, you may separate your 150-200 years evaluation period into 30-year time slices and check whether the impact of the lake depth change is a robust feature or has a larger natural variability.*

**Thanks a lot for your comment. In the $\delta_{depth}$ experiments the lake depth, as well as the lake and vegetation extent are kept constant throughout the mH and mHL10 simulations. Therefore, the results only show the effect of the constant differences in lake depth on the climate (and not of the variability in the lake depth). Since the natural variability of monsoon precipitation in the Sahel is high, an averaging period of 150 years is required to obtain robust results (so that the precipitation response is not overshadowed by the high natural variability).**
**We have clarified this statement in the relevant section (revised Manuscript: L382-391).**

*I suggest adding a table with regional averages (e.g. Sahel and Sahara) in precipitation for the different experiments that would allow an easy comparison of the different precipitation changes (see also my comment to Line 323-325).*

**We have provided the regional mean annual precipitation and standard deviation of the annual precipitation for the Sahel, Sahara and the individual basins in a table in the appendix.**

*I also suggest a thorough proof reading as the current version of the manuscript comprises several typos. Below I suggested correcting those I found.*

**We have corrected the typos as suggested.**

*In summary, I suggest accepting the paper for publication after minor revisions are conducted.*

**Minor remarks**

In the following suggestions for editorial corrections are marked in Italic.

*Line 11: … Basin, the lake area is slightly …*

**Ok.**

*Line 16: … lake expansion that is dominated by the expansion …*

**Ok.**

*Line 16-18: Sentence is somewhat difficult to read. Please rephrase, e.g. separate into two sentences!*

**We have rephrased this sentence as the follwoing: "Accordingly, the surface temperature increases over the region of Lake Chad and causes local changes in meridional surface-temperature gradient. These changes in the meridional surface-temperature gradient are associated with a reduced inland moisture transport from the tropical Atlantic into the Sahel, which causes a drying response in the Sahel."**

*Line 54: … treated, all previous simulation studies prescribe …*

*This statement sems to be too general. I guess you mean studies with GCMs? With regard to hydrology, there are several studies that use climate forcing to simulate lake area expansion in the mid-Holocene, e.g. Coe 1997: Stacke 2011.*

*Coe, M.: Simulating continental surface waters: An application to holocene Northern Africa, J. Climate, 10, 1680–1689, 1997.*

*Stacke, T. (2011). Development of a dynamical wetlands hydrology scheme and its application under different climate conditions. Phd Thesis, Hamburg: University of Hamburg. Berichte zur Erdsystemforschung, 99.*

**Yes, this statement seemed to be in fact too general or even misleading and therefore needed to be rephrased. We here wanted to refer to studies that investigated the effect of North African lakes on the mid-Holocene climate using GCM only. We have therefore rephrased the sentence as the following: "Apart from the difference in how the vegetation is treated, previous simulation studies all prescribe the mid-Holocene lake extent over the Sahel and Sahara from reconstructions *to investigate there effect on the mid-Holocene climate* (Li et al., 2023; Specht et al., 2022; Chandan and Peltier, 2020; Krinner et al., 2012; Broström et al., 1998; Coe and Bonan, 1997)."**

**We think the suggested citations (Coe 1997; Stacke 2011) would rather contribute to the statement at line 38-41, since these studies focus on modeling the mid-Holocene surface water extent using a prescribed mid-Holocene climate/hydrological forcing. We have therefore included the citation in the previous paragraph.**

*Line 69: In the end of sect. 1, an outline about the following sections is missing. The last sentence only describes the content of Sect. 2.*

**We are sorry for the missing outline. We corrected the numbering of the individual sections as follows: 1. introduction, 2. methods, 3. results and 4. conclusions. At the end of the introduction, we added a brief overview on the following sections: "In the following section 2,**

**we describe the concept of the dynamic lake model and the structure of the present-day and mid-Holocene simulations. In section 3, we evaluate the simulated present-day lake extent by comparing it with observational data. We compare the simulated mid-Holocene precipitation increase and the lake and vegetation extent with mid-Holocene reconstruction data. Additionally, we analyse the individual and synergistic effect of the mid-Holocene lake and vegetation extent on the North African climate and how changes in the lake depth influence the mid-Holocene climate over North Africa. Finally, we discuss our results in relation to former studies in the conclusions (section 4)."**

*Line 91: …applied over northern Africa.*

**Ok.**

*Line 142: … concept of the endorheic …*

**Ok.**

*Line 214: … represented as a mixed …*

**Ok.**

*Line 228: …orography is used …*

**Ok.**

*Line 242: … less sensitive to …*

**Ok.**

*Line 245: … growth and shrinking of …*

**Ok.**

*p.12 – Figure 5 caption – last sentence: It is written: "The black boundary in subplot c) …" I assume you mean panel d) not c)? In addition, I suggest writing 'panel' instead of 'subplot' throughout the paper.*

**Yes, we meant panel d), not c). Sorry for the confusion. We replaced "subplot" with "panel" throughout the document as suggested.**

*Line 272: Considering these large …*

**Ok.**

*Line 313: … presence of dynamic lakes in the ICON-JSBACH4 …*

**Ok.**

*Line 324: … to the total …*

**Ok.**

*Line 323-325: I suggest providing some values (e.g. averaged over the Sahel) to allow an easy comparison of the precipitation changes.*

**We provided these information in an extra table A1 in the appendix.**

*p. 16 – Fig. 8 caption: Simulated a) lake extent changes …*

**Ok.**

*Line 333: In fact, the comparison …*

**Ok.**

*Line 340: … the different scaling).*

**Ok.**

*Line 355 and 357: There are no arrows in Fig. 9c. Please correct!*

**We realized that Panel b) is systematically referred to as Panel c) by mistake. We corrected this error.**

*Line 373-376: Sentence is too long and difficult to read. Please rephrase into two sentences.*

**We agree that this sentence is hard to understand. Therefore, we rephrased it as the following: „This circulation response includes a near-surface easterly wind acceleration above Lake Chad (Fig. 8d) that decreases the inland moisture transport and, thus, rainfall at around 12 ◦N (Fig. 9b). Additionally, the circulation response includes a dipole-like zonal wind response in the mid-troposphere above Lake Chad that corresponds to a southward shift of the African Easterly Jet (Fig. 8d) and, thus, a southward shift of the rain belt's northern boundary."**

*Line 385: However, the simulated …*

**Ok.**

*Line 391: …contributes a large …*

**Ok.**

*Line 402: … updrafts, which occurs …*

**Ok.**

*Line 406-408: Sentence is difficult to read and understand. Please rephrase!*

**We rephrased the sentence accordingly: "Reconstructions show that aquifers over North Africa filled up in the early Holocene until the water table reached the overlying lake basins, leading to the formation of larger lakes (Lezine et al., 2011b). As precipitation over northern Africa decreased towards the end of the African Humid Period, the lake basins continued to be fed by the aquifers (Lezine et al., 2011b). The expansion and regression of the lakes, therefore, occurred with a delay of about 3,000-year compared to the orbital-forced summer insolation changes (Lezine et al., 2011b)."**

*Line 417: …relates to the known dry …*

**Ok.**

*Line 447-457: This paragraph comprises the same or similar sequence of references several times. Please rephrase and avoid redundant use of the same references if possible.*

**We rephrased this paragraph as the following: „Unfortunately, previous mid-Holocene simulation studies provide only little information on how lakes are represented in the utilized climate models in terms of the lake depth or the lake surface albedo (Coe and Bonan, 1997; Brostrom et al., 1998; Carrington et al., 2001; Krinner et al., 2012; Chandan and Peltier,**

**2020; Li et al., 2023). These previous simulation studies likely neglect the effect of the mid-Holocene lake depth changes on the monsoon precipitation because they use lake reconstructions that rarely provide information about the lake depth (e.g. Hoelzmann et al., 1998). The differences in the lake representation between different climate models might be the reason why in some models, a prescribed mid-Holocene lake extent causes a local and marginal precipitation increase only (Coe and Bonan, 1997; Brostrom et al., 1998; Chandan and Peltier, 2020), whereas the same lake extent causes a substantial precipitation increase across northern Africa in other models (Krinner et al., 2012; Specht et al., 2022; Li et al., 2023). In our study, lakes are treated as a pure mixed layer that has a dynamic depth and a constant surface albedo of 0.07. A more realistic lake surface temperature might be simulated by taking into account the existence of a lake thermocline and a dynamic lake albedo."**

*Line 465: … over northern Africa.*

**Ok.**

*Line 466: … effect of exorheic …*

**Ok.**